

# A correspondence between the Rabi model and an Ising model with long-range interactions

Bruno Scheihing-Hitschfeld[1⋆] and Néstor Sepúlveda[2†]

**1** Kavli Institute for Theoretical Physics, University of California, Santa Barbara, California 93106, USA
**2** School of Engineering and Sciences, Universidad Adolfo Ibáñez, Diagonal las Torres 2640, Peñalolen, Santiago, Chile

⋆ bscheihi@kitp.ucsb.edu , † nesepulv@gmail.com

## Abstract

By means of Trotter's formula, we show that transition amplitudes between a class of generalized coherent states in the Rabi model can be understood in terms of a certain Ising model featuring long-range interactions beyond nearest neighbors in its thermodynamic limit. Specifically, we relate the transition amplitudes in the Rabi model to a sum over binary variables of the form of a partition function of an Ising model with a number of spin sites equal to the number of steps in Trotter's formula applied to the real-time evolution of the Rabi model. From this, we show that a perturbative expansion in the energy splitting of the two-level subsystem in the Rabi model is equivalent to an expansion in the number of spin domains in the Ising model. We conclude by discussing how calculations in one model give nontrivial information about the other model, and vice versa, as well as applications and generalizations this correspondence may find.


# 1   Introduction

When two apparently distinct physical theories can be shown to be equivalent, there is an opportunity to understand more deeply the inner workings of both theories, especially when the regimes in which each theory is tractable correspond to values of the parameters/couplings that are intractable from the point of view of the other theory. Fundamental dualities across different areas of physics reveal deep connections between seemingly distinct frameworks. In quantum mechanics, the Fourier transform formalizes the position-momentum duality [1, 2], showing that a system's description in position space is mathematically equivalent to its representation in momentum space, directly tied to the wave-particle duality [3–5], where quantum entities exhibit both wave-like and particle-like behavior depending on observation. In statistical mechanics, the Kramers-Wannier duality [6, 7] in the Ising model relates high-temperature and low-temperature regimes, offering insights into phase transitions. In condensed matter and quantum field theory, bosonization [8–11] establishes an equivalence between fermionic and bosonic descriptions in (1+1) dimensions, simplifying the study of interacting fermions. At the interface of gravity and quantum field theory, the AdS/CFT correspondence [12–14] provides a striking example of holographic duality, equating a higher-dimensional gravitational theory to a conformal field theory on its lower-dimensional boundary, offering profound insights into quantum gravity and strongly coupled systems. These dualities not only facilitate calculations but also deepen our conceptual understanding of physical phenomena.

   In all of these examples, it is possible to formulate physical questions in one theory and answer them using the machinery of the other, equivalent description. Some observables that may be very difficult to calculate in one description may be simple in the other, and vice-versa.

   In this work, by means of Trotter's formula [15] applied to real-time quantum evolution, we show that there exists a correspondence between the quantum mechanical transition amplitudes of the Rabi model [16,17] and the partition function of a specific 1D Ising model [18] with nonlocal interactions. That is to say, we show that the dynamics of a quantum theory may be understood in terms of the partition function of a higher-dimensional theory with a different set of degrees of freedom, and vice-versa. We also show how to cast this correspondence in imaginary time, allowing for well-developed methods in statistical physics to have direct applicability.

   On the one hand, the Rabi model is a theory of interest as a model of the interaction of electromagnetic waves with an atom in a cavity. In a unit system where $\hbar = 1$, the Hamiltonian can be expressed as

$$\mathcal{H} = \omega_0 \sigma^z + \omega a^\dagger a + g \sigma^x (a + a^\dagger), \tag{1}$$

where $a$, $a^\dagger$ represent the annihilation and creation operators of a photon of frequency $\omega$, and

$\sigma^k$ with $k \in \{x, y, z\}$ are the Pauli matrices, modeling the atom in the cavity as a two-level system with an energy splitting of $\omega_0$. Lastly, the term proportional to $g$ describes interactions between the two sectors of the theory.

The Rabi model has been extensively studied across various regimes. It is often solved using the Rotating Wave Approximation (RWA), which is valid near resonance when counter-rotating terms oscillate much faster than rotating terms and can be neglected, leading to the well-known Jaynes-Cummings model [19]. This simplification enables analytical solutions and experimental confirmation revealing quantum effects such as collapse and revival of atomic inversion [20, 21], spontaneous emission, and absorption [19, 22].

However, the RWA's validity depends on the coupling ratio $g/\omega$. While it holds in the strong coupling regime ($g/\omega < 0.1$), it breaks down in the ultra-strong ($g/\omega \geq 0.1$) and deep strong coupling ($g/\omega > 1$) regimes, even at exact resonance [23–28].

While these extreme regimes are challenging to achieve in quantum-optical cavity quantum electrodynamics (QED), circuit QED offers tunable couplings that allow counter-rotating terms to generate novel quantum effects beyond the predictions of the RWA [23–25]. These effects include ultrastrong coupling state engineering and tomography [29], sub-cycle switching of ultrastrong light-matter interactions [30], Bloch-Siegert frequency shifts [31, 32], and violations of selection rules leading to unconventional transitions [33].

From a theoretical perspective, solutions for its energy levels and eigenfunctions have been obtained in terms of transcendental functions related to Heun functions [34]. And while much is understood about the theory, it is not too difficult to encounter quantities for which more tools would be desirable. Consider preparing the photons in a "classical" coherent state $|\alpha\rangle_c$, characterized by $a|\alpha\rangle_c = \alpha|\alpha\rangle_c$. A natural question to ask is how this state evolves when interacting with the two-level system via Eq. (1), which we can characterize by calculating

$$\mathcal{A}_{s'\beta,s\alpha} = \left\langle s', \beta \right|_c e^{-i\mathcal{H}t} |s, \alpha\rangle_c = \left\langle s', \beta \right|_c U(t) |s, \alpha\rangle_c \,, \tag{2}$$

where $U(t)$ represents the time evolution operator and $s, s'$ the eigenvalues of $\sigma^z$.

In this work, we will derive a formula to calculate a family of directly related amplitudes – which may in fact be used to reconstruct Eq. (2) – without solving a differential equation. We will do so in terms of a set of generalized coherent states, defined as eigenstates of spin-dependent annihilation operators $b \equiv \sigma^x a$ and a parity operator $\Pi$, introduced later in the text. For brevity, we will denote these generalized states simply by $|\alpha\rangle$ by working in a fixed parity sector, with the understanding that they differ from the usual coherent states of $a$ (denoted, as above, by $|\alpha\rangle_c$). Explicit formulas and further details of these states are provided in Appendix A, where the parity operator $\Pi$ and the photon-atom entanglement structure of these states are discussed at length.

The formula we will derive is nothing else than the partition function of an Ising model.

The Ising model is one of the oldest and most well-studied systems in physics. It consists of a chain of spins, each taking values of $+1$ or, $-1$ which interact with their nearest neighbors. The system evolves to minimize its energy, often leading to an ordered (ferromagnetic) or disordered (paramagnetic) phase, depending on temperature and external fields. However, despite of its ubiquitousness, these spin chains can be as complex as the interactions between the individual sites.

The form that we will consider in this work is

$$\mathcal{H}_{\text{Ising}} = -\sum_{i,j} \sigma_i^z K_{i,j} \sigma_j^z - \sum_i \sigma_i^z B_i \,, \tag{3}$$

where $\sigma_i^z$ is the Pauli matrix in the $z$-direction representing a spin at site $i$, $K_{i,j}$ represents the interactions between spins, and $B_i$ a local magnetic field that favors one orientation over the other. From a quantum mechanical point of view, there is nothing complicated about

this model: its eigenstates are simply products of the eigenstates of each Pauli matrix $\sigma_i^z$, and the energy of each state is obtained by evaluating the sum in Eq. (3) for a given spin configuration. However, any calculation with it will, in principle, be as complicated as the interaction matrix $K_{i,j}$. One quantity of interest in such a model is its partition function, as it encodes the thermodynamic properties of the spin chain. Taking the trace over the basis of eigenstates (for simplicity, in units where $k_B T = 1$), one obtains

$$Z = \sum_{\{s_i\}_{i=1}^N} \exp\left( \frac{1}{2} \sum_{i,j} s_i K_{i,j} s_j + \sum_i s_i B_i \right), \tag{4}$$

for which there is no simple, closed expression unless more information is given on $K$ and $B$ such that the sum may be carried out. We will show that a sum of this form determines the value of the amplitude (8).

In what follows, two steps are necessary to make this connection: to recast the annihilation and creation operators $a, a^\dagger$ in terms of $b \equiv \sigma^x a$ and $b^\dagger \equiv \sigma^x a^\dagger$. The advantage of doing this is that the Rabi Hamiltonian takes the form

$$\mathcal{H} = -\omega_0 e^{i\pi b^\dagger b} \Pi + \omega b^\dagger b + g(b^\dagger + b), \tag{5}$$

where $\Pi = -e^{i\pi a^\dagger a}\sigma^z = -e^{i\pi b^\dagger b}\sigma^z$ is a parity operator, i.e., it has discrete eigenvalues $P = \pm 1$, and it commutes with $\mathcal{H}$, meaning that we may simply replace $\Pi$ by its eigenvalue when studying the dynamics of the system. By doing this, we have reduced the problem to one set of creation and annihilation operators, satisfying the usual commutation relations $[b, b^\dagger] = 1$, and whose action on a state do not mix different parity subsectors: $\Pi b \Pi = b$. The eigenstates of $a$ and $b$ are nonetheless related to each other by taking appropriate linear combinations. For example,

$$|P = -1, \alpha\rangle = \frac{1}{2}\left( |+, \alpha\rangle_c + |+, -\alpha\rangle_c + |-, \alpha\rangle_c - |-, -\alpha\rangle_c \right), \tag{6}$$

$$|+, \alpha\rangle_c = \frac{1}{2}\left( |P = -1, \alpha\rangle + |P = -1, -\alpha\rangle + |P = +1, \alpha\rangle - |P = +1, -\alpha\rangle \right). \tag{7}$$

That is to say, the amplitudes expressed in the $b$ basis can be fully re-expressed in terms of the original $a$ operators by superimposing states from the two parity operator $\Pi$ eigenvector subspaces. Eqs. (6) and (7) are two examples of how the coherent states defined by $b = \sigma^x a$ can be written as linear combinations of $a$-coherent states with fixed spin states and vice-versa. The rest of these relations are listed in Appendix A. These identities demonstrate explicitly that knowledge of overlaps in the $b$ basis is equivalent to knowledge of overlaps in the $a$ basis.

Therefore, our object of interest in this work, which fully characterizes (albeit indirectly) the amplitude in Eq. (2), is

$$\mathcal{A}_{\beta\alpha} = \langle \beta | e^{-i\mathcal{H}t} | \alpha \rangle = \langle \beta | U(t) | \alpha \rangle, \tag{8}$$

where we have suppressed the parity label, as we do throughout the rest of the text.

## 2  Trotter's formula and the real-time correspondence

The other ingredient we need in order to make this connection is Trotter's formula [15]. Let's assume we start with an arbitrary generalized coherent state $|\alpha\rangle$, characterized by $b|\alpha\rangle = \alpha|\alpha\rangle$. To find the evolution of this state in time, let us define

$$\mathcal{H} = \mathcal{H}_1 + \mathcal{H}_2 + \mathcal{H}_3, \tag{9}$$

with $\mathcal{H}_1 = -P\omega_0 e^{i\pi b^\dagger b}$, $\mathcal{H}_2 = \omega b^\dagger b$ and $\mathcal{H}_3 = g(b^\dagger + b)$. Finding the action of $U(t)$ is not trivial since $\mathcal{H}_1$ and $\mathcal{H}_2$ do not commute with $\mathcal{H}_3$. However, we can use Trotter's formula

$$U(t) = \lim_{n\to\infty} \left( e^{-i\mathcal{H}_1 t/n} e^{-i\mathcal{H}_2 t/n} e^{-i\mathcal{H}_3 t/n} \right)^n , \tag{10}$$

to make progress.

Let $U_n$ be defined as

$$U_n = e^{-i\lambda_n b^\dagger b} e^{-i\delta_n e^{i\pi b^\dagger b}} e^{\gamma_n(b^\dagger + b)} , \tag{11}$$

where $\lambda_n = \omega t/n$, $\delta_n = -P\omega_0 t/n$ and $\gamma_n = -igt/n$. One can then apply $U_n$ $n$ times over the state $|\alpha\rangle$ to calculate its time evolution. It follows that

$$e^{\gamma_n(b^\dagger + b)} |\alpha\rangle = e^{\gamma_n \mathrm{Re}(\alpha)} |\alpha + \gamma_n\rangle , \tag{12}$$

$$e^{-i\delta_n e^{i\pi b^\dagger b}} |\alpha\rangle = \cos\delta_n |\alpha\rangle - i\sin\delta_n |-\alpha\rangle , \tag{13}$$

$$e^{-i\lambda_n b^\dagger b} |\alpha\rangle = \left| e^{-i\lambda_n}\alpha \right\rangle , \tag{14}$$

and therefore, that

$$U_n |\alpha\rangle = e^{\gamma_n \mathrm{Re}(\alpha)} \left[ \cos\delta_n \left| (\alpha + \gamma_n)e^{-i\lambda_n} \right\rangle - i\sin\delta_n \left| -(\alpha + \gamma_n)e^{-i\lambda_n} \right\rangle \right] . \tag{15}$$

That is to say, each step in the Trotterized evolution of the state includes a continuous action on the state plus a splitting mediated by the interaction proportional to $\omega_0$.

We can then consider the transition amplitudes between coherent states. Using Trotter's formula in Eq. (8), these are given by

$$\mathcal{A}_{\beta\alpha} = \langle\beta| U(t) |\alpha\rangle = \lim_{n\to\infty} \langle\beta| U_n^n |\alpha\rangle . \tag{16}$$

From the structure of the previous equation it is clear that the Trotterized amplitude $\langle\beta| U_n^n |\alpha\rangle$ at a fixed $n$ will contain $2^n$ terms. Furthermore, each action of $U_n$ modifies the components of the state by flipping the sign of the eigenvalue of $b$. It should therefore not come as a surprise that this sum of $2^n$ terms may be written in terms of a sum over the possible configurations of $n$ sign variables $s_i \in \{-1, 1\}$, with $i = 1, \ldots, n$. That is to say, a sum like the one that appears in the partition function of the Ising model.

Indeed, using results from Appendices B and C, in Appendix D we show that

$$\lim_{n\to\infty} \langle\beta| U_n^n |\alpha\rangle = \lim_{n\to\infty} \left( \frac{-i\sin(2\delta_n)}{2} \right)^{n/2} e^{-\frac{|\alpha|^2 + |\beta|^2}{2}} \tag{17}$$

$$\times \left\{ \sum_{\substack{\{s_k\}_{k=1}^{n-1} \\ s_0 = s_n = 1}} e^{\frac{1}{2}\sum_{j,\ell=1}^{n-1} s_j s_\ell K_{j,\ell} + \sum_{\ell=1}^{n-1} s_\ell B_\ell^+ + e^{-i\omega t}\beta^*\alpha} + \sum_{\substack{\{s_k\}_{k=1}^{n-1} \\ s_0 = -s_n = 1}} e^{\frac{1}{2}\sum_{j,\ell=1}^{n-1} s_j s_\ell K_{j,\ell} + \sum_{\ell=1}^{n-1} s_\ell B_\ell^- - e^{-i\omega t}\beta^*\alpha} \right\},$$

where

$$K_{j,\ell} = \delta_{\ell,j+1} \ln(i\cot\delta_n) + \gamma_n^2 e^{-i|j-\ell|\lambda_n} , \tag{18}$$

$$B_\ell^\pm = \gamma_n \left[ \beta^* e^{-i\ell\lambda_n} \pm \alpha e^{-i(n-\ell)\lambda_n} \right] . \tag{19}$$

Note that $\delta_{\ell,j+1}$ in Eq. (18) denotes a Kronecker delta, ensuring that the logarithmic term contributes only when $\ell = j + 1$. These expressions have exactly the structure of the partition function of an Ising model, with the important feature that several quantities are complex (as is necessary for this formulation to reproduce a quantum mechanical amplitude). This is our first main result.

In practice, calculating these amplitudes directly in the Rabi model becomes increasingly expensive at large values of $\alpha$ and $\beta$, since one must numerically represent a Hilbert space large enough to accommodate the coherent states $|\alpha\rangle$ and $|\beta\rangle$. Exact diagonalization methods require a cutoff $N_{\max}$ on the photon number basis, with $N_{\max}$ typically scaling as $|\alpha|^2$ to achieve convergence, because the Poissonian weight $p_N = e^{-|\alpha|^2}|\alpha|^{2N}/N!$ extends up to $N \approx |\alpha|^2$. By contrast, our mapping avoids this: it relies on the partition function of an associated Ising model and thus does not require explicit overlaps with a truncated bosonic basis. In our approach, the cost is completely contained within the time evolution. In particular, the Trotter discretization converges with a number of time slices $n$ that scales linearly with $|\alpha|$ (see Appendix B.2). As we discuss in the next section in terms of an expansion in the number of "domain walls" of the partition function, and in more generally in Appendix B.2, this can lead to more efficient calculations than the Hilbert space truncation.

We close this section by noting that this expression may be rewritten in terms of imaginary time $\tau = it$ to make the closest connection with well-developed methods of statistical physics. We discuss this further in Section 4.

## 3 The continuum limit of the Ising model as a perturbative expansion in $\omega_0$

Given that the full amplitude $\langle\beta|U(t)|\alpha\rangle$ in Eq (17) is given by a limit, it is natural to ask if there is an explicit way to take the limit. Viewed as a spin chain, all of the terms in Eqs. (18) and (19) have a straightforward continuum limit because the $1/n$ factor in $\gamma_n$ converts the sums into Riemann sums that converge to integrals, except for the nearest neighbor terms controlled by $\delta_n$.

In the continuum limit $n \to \infty$, the magnitude $\ln|\cot\delta_n|$ grows without bound, meaning it is energetically favorable to have as few sign flips as possible. It then becomes natural to organize the sum over the sign variables as a function of the number of sign flips in the sequence $\{s_i\}$. Let $m$ be that number. In the continuum limit, the position of the individual sign flips $\{i_k\}_{k=1}^m$ may be characterized by uniform random variables $z_k \in [0,1]$, each one corresponding to the value of $i_k/n$.

Following through with this rearrangement, we may take the limit $n \to \infty$ at each fixed value of $m$ and obtain a series expression for $\langle\beta|U(t)|\alpha\rangle$. We show details of this derivation in Appendix D. We obtain

$$\langle\beta|U(t)|\alpha\rangle = e^{-\frac{|\alpha|^2 - 2\beta^*\alpha + |\beta|^2}{2}} e^{\frac{ig^2 t}{\omega}} \sum_{m=0}^{\infty} \frac{(iP\omega_0 t)^m}{m!} e^{-\frac{2mg^2}{\omega^2} - (\alpha + \frac{g}{\omega})(\beta^* + \frac{g}{\omega})[1 - (-1)^m e^{-i\omega t}]} F_m, \quad (20)$$

where $F_m$ is given by

$$F_m = \left\langle \exp\left( -\frac{4g^2}{\omega^2} \sum_{k=1}^m \sum_{\ell=1}^{k-1} (-1)^{k+\ell} e^{-i(z_k - z_\ell)\omega t} - \frac{2g}{\omega} \sum_{k=1}^m \left[ \left(\beta^* + \frac{g}{\omega}\right)(-1)^k e^{-iz_k \omega t} \right.\right.\right. \quad (21)$$

$$\left.\left.\left. - \left(\alpha + \frac{g}{\omega}\right)(-1)^{m+k} e^{-i(1-z_k)\omega t} \right] \right) \right\rangle_m .$$

The average $\langle\cdot\rangle_m$ is taken over the possible sign flip positions $z_k \in (0,1)$ (the "domain wall" positions in the spin chain), with $\{z_k\}_{k=1}^m$ an ordered sequence. Explicitly,

$$\langle G\rangle_m = m! \int_0^1 dz_m \int_0^{z_m} dz_{m-1} \ldots \int_0^{z_2} dz_1 G(z_1, z_2, \ldots, z_m). \quad (22)$$

Eq. (20) is our second main result.

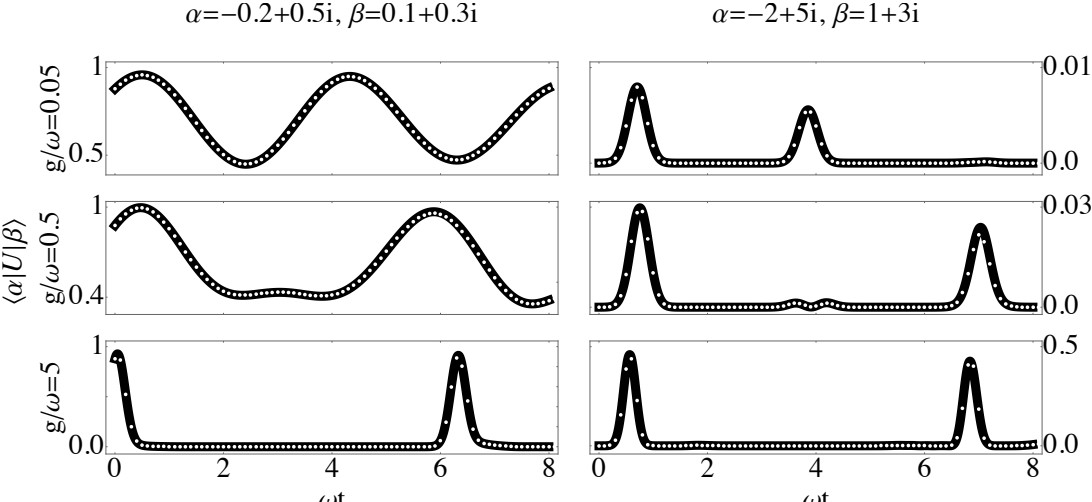

Figure 1: Transition amplitudes between two pairs of generic coherent states: $|\alpha\rangle = |-0.2 + 0.5i\rangle$ and $|\beta\rangle = |0.1 + 0.3i\rangle$ (first column), and $|\alpha\rangle = |-2 + 5i\rangle$ and $|\beta\rangle = |1 + 3i\rangle$ (second column), considering parity equal to $+1$. The black curves represent the numerical solution of the Schrödinger equation for the Rabi Hamiltonian, while the white dotted curves correspond to Eq. (20) with $m_{max} = 10$ and $10^4$ sampling steps for each value of $m$ from 1 to $m_{max}$ to compute the mean value and obtain $F_m$ using Eq. (21). For a given value of $m$, which represents the number of domain walls (or spin flips) in the spin chain Hamiltonian, a sampling step corresponds to the random assignment of spin flip positions along the chain, independently distributed within the interval $(0, 1)$. In all plots we used $\omega_0/\omega = 0.3$. We compare both results across three coupling regimes: strong coupling ($g/\omega = 0.05$, first row), ultra-strong coupling ($g/\omega = 0.5$, second row), and deep strong coupling ($g/\omega = 5$, third row).

It is clear from Eq. (20) that the result of taking the continuum limit in this form yields a perturbative expansion in $\omega_0$, of the same form that one would get using time-dependent perturbation theory. While on the one hand the limit $n \to \infty$ has been taken, the convergence of the series requires more and more terms as $t$ grows larger, and each coefficient of the series (which are also time-dependent) requires one to evaluate averages for which we have not found a closed form, meaning that the practical applicability of this formula is not obviously superior to the one given earlier in Eq. (17). However, it provides one with another, independent handle to calculate the transition amplitudes.

Figure 1 presents a comparison of the numerical solution of the Schrödinger equation for the Rabi Hamiltonian with Eq. (20) across the three previously mentioned regimes: strong coupling, ultra-strong coupling, and deep strong coupling. It is important to emphasize that Eq. (20) was derived from Eq. (17), and that the interpretation in terms of a Ising model is still transparent: $m$ is the number of "domain walls" (or sign flips) in the spin chain Hamiltonian defined by Eq. (17).

It is worth emphasizing that in going from Eq. (17) to Eq. (20), $m$ is always smaller than $n$, meaning that the information encoded in the first $n$ terms of Eq. (17) is contained within the first $n$ terms of Eq. (20). Taken together with the estimate in Appendix B.2, this suggests that to stay below an error tolerance $\varepsilon$, one needs $m \gtrsim C_1 g \omega |\alpha| t^2/\varepsilon$, scaling linearly with $\alpha$. However, inspecting Eq. (20), we see that this might be an overly conservative estimate, as the result is organized as a power series in $\omega_0 t$, while the sensitivity to $\alpha, \beta$ is purely contained

in the integral moments $F_m$. In particular, a scan over the values of $\omega_0$ could be done (up to a given tolerance fixed by the value of $\omega_0 t$) at significantly lower cost than with a direct diagonalization approach if one calculates the functions $F_m$ in advance.

All the results presented so far correspond to the real-time dynamics of the Rabi model. The derivation based on Trotter's formula and the mapping to the effective Ising chain, as well as the continuum-limit expansion in Eq. (20), have all been carried out in real time, preserving the unitary time evolution of the quantum system. This distinguishes our approach from the usual imaginary-time (Euclidean) formulations, which are typically used to study equilibrium properties. In contrast, our construction explicitly captures the coherent dynamics and interference effects, such as the revival phenomena observed in Fig. 1, which are intrinsic features of the real-time quantum evolution.

## 4 Applications in imaginary time

We now look at likely applications of this correspondence beyond the calculation of amplitudes just described, first starting from calculations on the Ising model side, and then commenting on the converse direction. We focus on imaginary time applications, as real partition functions are natural objects to calculate from the Ising side of the correspondence.

### 4.1 What this Ising model can do for the Rabi model

We begin by considering the imaginary time version of the amplitudes $\langle \beta | U(t) | \alpha \rangle$. It is not hard to see that upon substituting $\tau = it$, the formula for the amplitudes is given by

$$\langle \beta | U | \alpha \rangle = \lim_{n \to \infty} \left( \frac{\sinh(2P\omega_0\tau/n)}{2} \right)^{n/2} e^{-\frac{|\alpha|^2+|\beta|^2}{2}} \tag{23}$$

$$\times \left\{ \sum_{\substack{\{s_k\}_{k=1}^{n-1} \\ s_0=s_n=1}} e^{\frac{1}{2}\sum_{j,\ell=1}^{n-1} s_j s_\ell K_{j,\ell}^E + \sum_{\ell=1}^{n-1} s_\ell B_\ell^{E+} + e^{-\omega\tau}\beta^*\alpha} + \sum_{\substack{\{s_k\}_{k=1}^{n-1} \\ s_0=-s_n=1}} e^{\frac{1}{2}\sum_{j,\ell=1}^{n-1} s_j s_\ell K_{j,\ell}^E + \sum_{\ell=1}^{n-1} s_\ell B_\ell^{E-} - e^{-\omega\tau}\beta^*\alpha} \right\},$$

where

$$K_{j,\ell}^E = \delta_{\ell,j+1} \ln\left( \coth \frac{P\omega_0\tau}{n} \right) + \frac{g^2\tau^2}{n^2} e^{-|j-\ell|\omega\tau/n}, \tag{24}$$

$$B_\ell^{E\pm} = -\frac{g\tau}{n} \left[ \beta^* e^{-\ell\omega\tau/n} \pm \alpha e^{-(n-\ell)\omega\tau/n} \right]. \tag{25}$$

It is instructive to analyze more closely the form of the Ising couplings in Eq. (24). The kernel $K_{j,\ell}^E$ has two contributions: (i) a nearest–neighbor term $\delta_{\ell,j+1} \ln\left( \coth(P\omega_0\tau/n) \right)$, which plays the role of a strong coupling between consecutive time slices in the Trotter decomposition, effectively enforcing that the spin configuration does not fluctuate arbitrarily from one slice to the next and thus depends on $\omega_0$, and (ii) a ferromagnetic, exponentially decaying tail $\frac{g^2\tau^2}{n^2} e^{-|j-\ell|\omega\tau/n}$ whose amplitude grows with $g^2$ and whose range is set by $\omega$. In the continuum limit, this second contribution becomes a nonlocal interaction kernel proportional to $e^{-\omega\tau|u-v|}$ with $u = j/n$, $v = \ell/n$. Taking the Fourier transform of this long-range part alone shows that its spectral weight has the form $\widetilde{K}(k) \propto [(\omega\tau)^2 + k^2]^{-1}$, i.e. a Yukawa-type kernel. This Fourier representation is not used in the subsequent calculations, but it is useful to highlight the effective range and structure of the induced interactions on the Ising side.

This structure reveals how different dynamical regimes of the Rabi model appear on the Ising side. For weak coupling $g \ll \omega, \omega_0$, the long–range piece is negligible and the dynamics is dominated by the short–range stiffness. Conversely, in the ultra-strong and deep-strong



coupling regimes $g \sim \omega$ or $g \gg \omega$, the ferromagnetic tail becomes long-ranged and sizable, strongly suppressing domain walls in imaginary time. This behavior encodes the crossover between perturbative dynamics and the strongly correlated regime, providing an explicit Ising–side diagnostic of the dynamical phases of the Rabi model.

The case where $\alpha = \beta = 0$ is a particularly simple instance of this formula, which we will revisit in the next section. In this case we simply have

$$\langle 0|U|0\rangle = \lim_{n\to\infty} \left( \frac{\sinh(2P\omega_0\tau/n)}{2} \right)^{n/2} \sum_{\substack{\{s_k\} \\ s_0=1}} e^{\frac{1}{2}\sum_{j,\ell=1}^{n-1} s_j s_\ell K_{j,\ell}^E}, \tag{26}$$

which is to say, it takes the form of an Ising model with long-range interactions in the absence of external fields.

Even though so far the only partition function we have discussed is that of the Ising model, starting from our expressions, it happens to be quite simple to derive a formula for the partition function of the Rabi model itself, which we will denote by $\mathcal{Z}$. Specifically, one can use the completeness relation

$$\mathbb{1} = \frac{1}{\pi} \int d^2\alpha |\alpha\rangle\langle\alpha|, \tag{27}$$

to calculate the trace of the time evolution operator

$$\frac{1}{2\pi} \int d^2\alpha \, \langle\alpha| U |\alpha\rangle = \sum_n e^{-E_n\tau} \equiv \mathcal{Z}, \tag{28}$$

where we identify the right hand side as the partition function of the Rabi Hamiltonian (in one of its parity subsectors) where $\tau = 1/T$ is the inverse temperature of the system. We emphasize that these imaginary-time amplitudes are not only of formal interest: They yield the finite-temperature partition function of the Rabi model (determined by the next three equations), from which physical quantities can be extracted as with any other partition function. It can also serve as a starting point for the calculation of correlation functions in imaginary time, which provides access to linear-response properties, making the amplitudes physically meaningful quantities beyond the mapping itself. Starting from Eq. (23), setting $\beta = \alpha$, carrying out the integral over $\alpha$ and proceeding as before to take the limit $n \to \infty$, one finds that

$$\mathcal{Z} = \sum_n e^{-E_n\tau} = \sum_{m=0}^{\infty} \frac{(\tau P\omega_0)^m}{m!} \frac{\mathcal{Z}_m}{1-(-1)^m e^{-\tau\omega}}, \tag{29}$$

where

$$\mathcal{Z}_m = \left\langle \exp\left( L[\bar{s}](\tau) + \frac{(-1)^m e^{-\tau\omega}(L[\bar{s}](\tau) + L[\bar{s}](-\tau))}{2(1-(-1)^m e^{-\tau\omega})} \right) \right\rangle_m, \tag{30}$$

and we have introduced

$$L[\bar{s}](\tau) = -\frac{4g^2}{\omega^2} \sum_{k=1}^{m}\sum_{\ell=1}^{k-1} (-1)^{k+\ell} e^{-(z_k-z_\ell)\omega\tau} - \frac{2g^2}{\omega^2}\sum_{k=1}^{m}(-1)^k \left[ e^{-z_k\omega\tau} - (-1)^m e^{-(1-z_k)\omega\tau} \right]$$
$$- \frac{g^2}{\omega^2}\left(1-(-1)^m e^{-\omega\tau}\right) - \frac{2mg^2}{\omega^2} + \frac{g^2\tau}{\omega}. \tag{31}$$

At large imaginary times $\tau \to \infty$ (corresponding to small temperatures), this expression is certainly more complicated than $e^{-E_0\tau}$. However, at moderate or small values of $\tau$, when all states contribute to the sum over eigenstates $\sum_n e^{-E_n\tau}$, Eq. (29) provides an explicit expression that avoids having to diagonalize a matrix of arbitrarily large dimension.

## 4.2   What the Rabi model can do for this Ising model

Although the Ising model obtained here is a specific instance thereof, with couplings determined by those of the Rabi model, we emphasize that partition functions like the ones we just encountered do appear in other contexts. For example, they share structural features with long-range interacting spin systems that naturally arise in cavity and waveguide QED [35]. The mapping therefore has relevance beyond a formal exercise, as it highlights a class of interactions of direct physical interest.

In particular, given that we know the Schrödinger equation that the quantum system satisfies, we can use it to obtain results for the partition function of the Ising model we described above. Concretely, the solution to the imaginary time Schrödinger equation

$$\partial_\tau |\psi\rangle = -\mathcal{H} |\psi\rangle \,, \tag{32}$$

with an initial condition specified by $|\psi(t=0)\rangle = |\alpha\rangle$, automatically returns the thermodynamic limit $n \to \infty$ of the partition function on the RHS of Eq. (23) by calculating its overlap at time $\tau$ with the state $\langle \beta |$.

If one knows the spectrum of $\mathcal{H}$ with enough precision to calculate the decomposition of a given generalized coherent state $|\alpha\rangle$ in terms of its eigenstates $|n\rangle$, then it is natural to write

$$\langle \beta | U | \alpha \rangle = \sum_n e^{-E_n \tau} \langle \beta | n \rangle \langle n | \alpha \rangle \,, \tag{33}$$

meaning that the sums in Eq. (23) may be decomposed in terms of a sum of decaying exponentials, thus providing an organizing principle to evaluate them at large values of $\tau$.

This last expression is most useful at values of $\alpha, \beta$ where only a small number of states $|n\rangle$ contribute to their decomposition in terms of eigenstates of $\mathcal{H}$. This will be the case, for example, when $g/\omega$ is small and $\alpha = \beta = 0$.

To close, we note that one relatively simple result of this correspondence is that the Ising model partition function we mentioned at the end of the last section can be written as

$$\lim_{n \to \infty} \left( \frac{\sinh(2P\omega_0 \tau/n)}{2} \right)^{n/2} \sum_{\substack{\{s_k\} \\ s_0 = 1}} e^{\frac{1}{2} \sum_{j,\ell=1}^{n-1} s_j s_\ell K_{j,\ell}^E} = \sum_n e^{-E_n \tau} |\langle 0 | n \rangle|^2 \,, \tag{34}$$

meaning that one can calculate the continuum limit of the spin system with no "magnetic fields" we described above by calculating the overlaps between the harmonic oscillator vacuum and the eigenstates of the Rabi Hamiltonian.

## 5   Conclusions and outlook

We have calculated the transition amplitudes between generalized coherent states (eigenstates of $b = a\sigma_x$ and the parity operator $\Pi$) in the Rabi model and found a remarkable correspondence with the partition functions of a family of Ising models with long-range interactions. These partition functions provide explicit expressions that converge to the Rabi model amplitudes when the continuum limit is taken. The key step in the derivation was to use Trotter's formula to study the dynamics of the Rabi model, by which the parameter $n$ that controls the number of steps in the Trotterized evolution ultimately becomes the number of sites in the "dual" Ising model. We have verified that our results reproduce the full quantum dynamics of the system, and discussed selected applications in which our expressions provide a more direct route to calculate a given observable than diagonalizing the Rabi Hamiltonian directly. This

perspective highlights the broader relevance of our approach and clarifies how the effective Ising couplings reflect the different dynamical regimes of the Rabi model.

An interesting next step would be to consider other quantum mechanical models with transition amplitudes (or other quantities) that admit a dual description in terms of a spin system. While it includes highly nonlocal interactions, the fact that we obtained an alternate way of calculating the partition function of a quantum mechanical system by evaluating a partition function of a spin system in one higher dimension is very reminiscent of dualities between quantum systems and gravitational systems (e.g., the matrix model dual to Jackiw-Teitelboim gravity [36]), although further study would be required to determine whether a notion of "bulk" and "boundary" can be established as in holographic dualities.

In principle, the present method could also be applied to the limit $\omega/\omega_0 \to 0$, where the Rabi model undergoes a superradiant phase transition characterized by the spontaneous buildup of a macroscopic photon field [37] (see also [38]). Within our Ising correspondence, this regime maps onto a situation where the effective long-range kernel $K_{j,\ell}^E$ strongly suppresses domain walls, favoring globally ordered spin configurations. We expect that in this spin representation, the onset of the superradiant phase would corresponds to the development of a nonzero magnetization — analogous to having a finite coherent amplitude in the bosonic description. Speculatively, the exponential decay of the kernel would suppress domain walls sufficiently to stabilize ordered phases such that it would mirror the emergence of superradiance in the Rabi model as $\omega/\omega_0 \to 0$. Although this lies beyond our present scope, our Ising formulation offers a framework to investigate this superradiant limit from a complementary angle.

Another avenue worth exploring using our results is the many-body physics of cavity and waveguide QED. Under various conditions, the dynamics of atom-light interactions can be described in terms of spin chains with long-range interactions [35], where their respective Hamiltonians are not at all unlike the Ising model expression we derived, as the long-range interactions in these spin chains are mediated by couplings $K_{i,j} \propto \exp(-\lambda|i-j|)$, where $\lambda$ may be imaginary or real and positive. The fact that this is exactly the form of the long-range interactions in the real- and imaginary-time versions of the coherent to coherent state transition amplitudes, Eq. (17) and Eq. (23), respectively, may open an interesting pathway to connect the dynamics of light coupled to a single atom with that of light interacting with a lattice of atoms, and therefore draw far-reaching lessons about the many-body physics of cavity and waveguide QED by studying a comparatively simpler quantum mechanical system.

## Acknowledgments

The authors sincerely thank Juan Carlos Retamal for highlighting the significance of the Rabi model, José Chesta-Lopez for insightful discussions during the early stages of this research, and Carla Hermann-Avigliano for providing experimental information on the three regimes of the Rabi model. The authors also thank the referees for their constructive comments and suggestions, which helped improve the clarity and scope of this work.

**Funding information** The work of BSH was supported in part by a CONICYT Grant Number CONICYT- PFCHA/MagísterNacional/2018-22181513, by grant NSF PHY-2309135 to the Kavli Institute for Theoretical Physics (KITP), and by grant 994312 from the Simons Foundation. NS was supported in part by Fondecyt grant No. 1220536 and the Millennium Science Initiative Program NCN2024_068 of ANID, Chile.

# A Hamiltonian model

We start with the Rabi model for the interaction of a quantum two-level atom with photons

$$\mathcal{H} = \omega_0 \sigma^z + \omega a^\dagger a + g(\sigma^+ + \sigma^-)(a^\dagger + a). \tag{A.1}$$

Here, the $\sigma$ operators are the Pauli Matrices acting on the two level system with basis $|+\rangle$ and $|-\rangle$, and $a^\dagger$ and $a$ are the creation and annihilation operators for the photon spectrum, with basis $|n\rangle$. The energy is tuned by the parameters $\omega_0$, $\omega$ and $g$, representing the energies of the atom levels gap, the photon frequency, and the interaction energy respectively. We have set $\hbar = 1$.

Expanding the interaction term we obtain

$$\mathcal{H} = \omega_0 \sigma^z + \omega a^\dagger a + g(\sigma^+ a + \sigma^- a^\dagger + \sigma^+ a^\dagger + \sigma^- a). \tag{A.2}$$

A common approach would be to neglect the counter-rotating terms, $\sigma^+ a^\dagger + \sigma^- a$, in favor of the rotating terms, $\sigma^+ a + \sigma^- a^\dagger$, by invoking the rotating wave approximation (RWA). Here we analyze the problem without employing this approximation.

We first begin by making a transformation to the Hamiltonian, mapping it into a photon system with a definite parity. Let the parity operator be

$$\Pi = -e^{i\pi a^\dagger a} \sigma^z. \tag{A.3}$$

If we take the basis $\{|s,n\rangle\}$, with $s = \pm 1$ representing the possible states of the atom, we see that the parity acts over this basis as

$$\Pi |s,n\rangle = -e^{i\pi a^\dagger a} \sigma^z |s,n\rangle = -s(-1)^n |s,n\rangle. \tag{A.4}$$

On the other hand, taking the target basis of the transformation $\{|P,n\rangle\}$, with $P = \pm 1$ representing the even and odd parities, we see that

$$\Pi |P,n\rangle = P |P,n\rangle. \tag{A.5}$$

We see that when the atom it's in the excited state ($s = 1$), the parity is odd when the photon number is even, and vice-versa. Conversely, when the atom it's in the ground state ($s = -1$), the parity is odd when the photon number is odd, and vice-versa. In conclusion, we can make a one-to-one correspondence between the $|s,n\rangle$ and the $|P,n\rangle$ bases.

With this transformation, the Hamiltonian is

$$\mathcal{H} = -\omega_0 e^{i\pi a^\dagger a} \Pi + \omega a^\dagger a + g(\sigma^+ + \sigma^-)(a^\dagger + a). \tag{A.6}$$

The elegancy of this transformation is that the system dynamics moves inside the Hilbert space split in two unconnected subspaces or parity chains

$$|-,0_a\rangle \leftrightarrow |+,1_a\rangle \leftrightarrow |-,2_a\rangle \leftrightarrow |+,3_a\rangle \leftrightarrow \ldots (P = +1), \tag{A.7}$$

$$|+,0_a\rangle \leftrightarrow |-,1_a\rangle \leftrightarrow |+,2_a\rangle \leftrightarrow |-,3_a\rangle \leftrightarrow \ldots (P = -1). \tag{A.8}$$

Neighboring states within each parity chain may be connected via either rotating or counter-rotating terms. For example, in the parity chain with $P = +1$, the counter-rotating term $\sigma^+ a^\dagger$ induces the transition $|-,2_a\rangle \rightarrow |+,3_a\rangle$, while the rotating term $\sigma^+ a$ induces $|+,1_a\rangle \leftarrow |-,2_a\rangle$.

The natural follow up question is if the basis $\{|P,n\rangle\}$ are eigenstates of the system. Clearly, since the interaction terms change the photon number the answer is no. We can see this explicitly

$$\mathcal{H} |P,n\rangle = [-\omega_0(-1)^n P + \omega n] |P,n\rangle + g\sqrt{n+1} |P,n+1\rangle + g\sqrt{n} |P,n-1\rangle. \tag{A.9}$$

Although the basis $\{|P, n\rangle\}$ is not an eigenenergy basis, we do conclude that the Hamiltonian is parity invariant. Therefore, in the following we will assume a fixed parity of the system and focus on only diagonalizing one of the disconnected Hilbert subspaces. With this assumption, the Hamiltonian is

$$\mathcal{H} = -\omega_0 e^{i\pi a^\dagger a} P + \omega a^\dagger a + g(\sigma^+ + \sigma^-)(a^\dagger + a). \tag{A.10}$$

Note that $\sigma^\pm, a, a^\dagger$ are defined on the full Hilbert space, not only on the fixed parity subspace. Given that we will focus on a fixed parity sector, and in order to eliminate this redundancy and simplify the calculation, we can make another canonical transformation to simplify the description of the system to a single set of creation and annihilation operators, without any spin variables remaining. Concretely, let $b = \sigma^x a$ and $b^\dagger = \sigma^x a^\dagger$. With this transformation $b^\dagger b = a^\dagger a$, therefore the photon number interpretation is not changed. Furthermore, the action of $b, b^\dagger$ does not take the state outside of the fixed parity subspace because $\Pi b \Pi = b$. The Hamiltonian follows as

$$\mathcal{H} = -\omega_0 e^{i\pi b^\dagger b} P + \omega b^\dagger b + g(b^\dagger + b). \tag{A.11}$$

Finally, we note that the coherent states with eigenvalue $\alpha$ associated with $b$ are related to the ones associated with $a$ via

$$|P = -1, \alpha\rangle = \frac{1}{2}\left(|+, \alpha\rangle_c + |+, -\alpha\rangle_c + |-, \alpha\rangle_c - |-, -\alpha\rangle_c\right), \tag{A.12}$$

$$|P = +1, \alpha\rangle = \frac{1}{2}\left(|-, \alpha\rangle_c + |-, -\alpha\rangle_c + |+, \alpha\rangle_c - |+, -\alpha\rangle_c\right), \tag{A.13}$$

$$|+, \alpha\rangle_c = \frac{1}{2}\left(|P = -1, \alpha\rangle + |P = -1, -\alpha\rangle + |P = +1, \alpha\rangle - |P = +1, -\alpha\rangle\right), \tag{A.14}$$

$$|-, \alpha\rangle_c = \frac{1}{2}\left(|P = +1, \alpha\rangle + |P = +1, -\alpha\rangle + |P = -1, \alpha\rangle - |P = -1, -\alpha\rangle\right). \tag{A.15}$$

## B  Trotterized time evolution of a coherent state

In the following section we will study the time evolution generated by the Hamiltonian in Eq. (A.11). To do this, we will consider a generalized coherent state

$$|\alpha\rangle_\pm = e^{-\frac{|\alpha|^2}{2}} \sum_{k=0}^{\infty} \frac{\alpha^k}{\sqrt{k!}} |P = \pm 1, k\rangle, \tag{B.1}$$

and to save space we will adopt the notation

$$|\alpha\rangle = e^{-\frac{|\alpha|^2}{2}} \sum_{k=0}^{\infty} \frac{\alpha^k}{\sqrt{k!}} |k\rangle. \tag{B.2}$$

Let's assume we start with an arbitrary coherent state $|\alpha\rangle$. The question to answer is: how does this state evolve in time? Therefore, we need to find the action of $U(t) = e^{-i\mathcal{H}t}$ over the state. Let the Hamiltonian be

$$\mathcal{H} = \mathcal{H}_1 + \mathcal{H}_2 + \mathcal{H}_3, \tag{B.3}$$

with $\mathcal{H}_1 = -\omega_0 e^{i\pi b^\dagger b} P$, $\mathcal{H}_2 = \omega b^\dagger b$ and $\mathcal{H}_3 = g(b^\dagger + b)$. Finding the action of $U(t)$ is not trivial since $\mathcal{H}_1$ and $\mathcal{H}_2$ do not commute with $\mathcal{H}_3$.

## B.1 Trotter's Formula

For $U(t) = e^{-i(\mathcal{H}_1 + \mathcal{H}_2 + \mathcal{H}_3)t}$ we can use Trotter's formula

$$U(t) = \lim_{n \to \infty} (e^{-i\mathcal{H}_1 t/n} e^{-i\mathcal{H}_2 t/n} e^{-i\mathcal{H}_3 t/n})^n . \tag{B.4}$$

Let $U_n$ be defined as

$$U_n = e^{-i\lambda_n b^\dagger b} e^{-i\delta_n e^{i\pi b^\dagger b}} e^{\gamma_n(b^\dagger + b)} , \tag{B.5}$$

where $\lambda_n = \omega t/n$, $\delta_n = -P\omega_0 t/n$ and $\gamma_n = -igt/n$. We will apply $U_n$, $n$ times over the state $|\alpha\rangle$ to calculate the time evolution. Then, the operator $D(\gamma_n) = e^{\gamma_n(b^\dagger + b)}$ is a displacement operator (since $\gamma_n$ is a pure imaginary) and we readily know its action over the state

$$e^{\gamma_n(b^\dagger + b)} |\alpha\rangle = D(\gamma_n)|\alpha\rangle = D(\gamma_n)D(\alpha)|0\rangle = D(\gamma_n + \alpha)e^{(\gamma_n\alpha^* - \gamma_n^*\alpha)/2}|0\rangle . \tag{B.6}$$

Since $\gamma_n$ is purely imaginary

$$e^{\gamma_n(b^\dagger + b)}|\alpha\rangle = e^{\gamma_n Re(\alpha)} D(\gamma_n + \alpha)|0\rangle = e^{\gamma_n Re(\alpha)}|\alpha + \gamma_n\rangle . \tag{B.7}$$

Next, we want to know the action of $e^{-i\delta_n e^{i\pi b^\dagger b}}$ over a state $|\alpha\rangle$

$$
\begin{aligned}
e^{-i\delta_n e^{i\pi b^\dagger b}} |\alpha\rangle &= e^{-|\alpha|^2/2} \sum_{k=0}^{\infty} \frac{\alpha^k}{\sqrt{k!}} e^{-i\delta_n e^{i\pi b^\dagger b}} |k\rangle \\
&= e^{-|\alpha|^2/2} \sum_{k=0}^{\infty} \frac{\alpha^k}{\sqrt{k!}} e^{-i\delta_n e^{i\pi k}} |k\rangle \\
&= e^{-|\alpha|^2/2} \sum_{k=0}^{\infty} \left( \frac{\alpha^{2k}}{\sqrt{(2k)!}} e^{-i\delta_n} |2k\rangle + \frac{\alpha^{2k+1}}{\sqrt{(2k+1)!}} e^{i\delta_n} |2k+1\rangle \right) .
\end{aligned}
\tag{B.8}
$$

Let's note that the coherent states $|\alpha\rangle$ and $|-\alpha\rangle$ can be expanded as:

$$|\alpha\rangle = e^{-|\alpha|^2/2} \left( \frac{\alpha^0}{\sqrt{0!}} |0\rangle + \frac{\alpha^1}{\sqrt{1!}} |1\rangle + \frac{\alpha^2}{\sqrt{2!}} |2\rangle + \frac{\alpha^3}{\sqrt{3!}} |3\rangle + \frac{\alpha^4}{\sqrt{4!}} |4\rangle + \cdots \right) , \tag{B.9}$$

$$|-\alpha\rangle = e^{-|\alpha|^2/2} \left( \frac{\alpha^0}{\sqrt{0!}} |0\rangle - \frac{\alpha^1}{\sqrt{1!}} |1\rangle + \frac{\alpha^2}{\sqrt{2!}} |2\rangle - \frac{\alpha^3}{\sqrt{3!}} |3\rangle + \frac{\alpha^4}{\sqrt{4!}} |4\rangle - \cdots \right) . \tag{B.10}$$

Therefore,

$$|\alpha\rangle + |-\alpha\rangle = 2e^{-|\alpha|^2/2} \sum_{k=0}^{\infty} \frac{\alpha^{2k}}{\sqrt{(2k)!}} |2k\rangle , \tag{B.11}$$

$$|\alpha\rangle - |-\alpha\rangle = 2e^{-|\alpha|^2/2} \sum_{k=0}^{\infty} \frac{\alpha^{(2k+1)}}{\sqrt{(2k+1)!}} |2k+1\rangle . \tag{B.12}$$

Using this we find the action of the operator

$$
\begin{aligned}
e^{-i\delta_n e^{i\pi b^\dagger b}} |\alpha\rangle &= \frac{1}{2} \left( e^{-i\delta_n}(|\alpha\rangle + |-\alpha\rangle) + e^{i\delta_n}(|\alpha\rangle - |-\alpha\rangle) \right) \\
&= \frac{e^{-i\delta_n} + e^{i\delta_n}}{2} |\alpha\rangle + \frac{e^{-i\delta_n} - e^{i\delta_n}}{2} |-\alpha\rangle \\
&= \cos\delta_n |\alpha\rangle - i\sin\delta_n |-\alpha\rangle .
\end{aligned}
\tag{B.13}
$$

Finally we wish to know the action of $e^{-i\lambda_n b^\dagger b}$ over a state $|\alpha\rangle$

$$e^{-i\lambda_n b^\dagger b}|\alpha\rangle = e^{-|\alpha|^2/2}\sum_{k=0}^{\infty}\frac{\alpha^k}{\sqrt{k!}}e^{-i\lambda_n b^\dagger b}|k\rangle = e^{-|\alpha|^2/2}\sum_{k=0}^{\infty}\frac{(e^{-i\lambda_n}\alpha)^k}{\sqrt{k!}}|k\rangle = \left|e^{-i\lambda_n}\alpha\right\rangle. \quad (B.14)$$

We now can conclude, summarizing our previous derivations that

$$U_n|\alpha\rangle = e^{\gamma_n Re(\alpha)}\Big[\cos\delta_n\left|(\alpha+\gamma_n)e^{-i\lambda_n}\right\rangle - i\sin\delta_n\left|-(\alpha+\gamma_n)e^{-i\lambda_n}\right\rangle\Big]. \quad (B.15)$$

With this, we see that the action of one unitary operator $U_n$ "splits" a coherent state into two pieces. For $n$ applications of $U_n$, we will be able to write the resulting state as a dot product between coherent states and coefficients:

$$U_n^n|\alpha\rangle = \begin{pmatrix} b_n^0 \\ b_n^1 \\ \vdots \\ b_n^{2^n-1} \end{pmatrix} \cdot \begin{pmatrix} \left|a_n^0\right\rangle \\ \left|a_n^1\right\rangle \\ \vdots \\ \left|a_n^{2^n-1}\right\rangle \end{pmatrix}. \quad (B.16)$$

We work this out explicitly in Appendix C, and we find that the dot product just written is explicitly determined by the functions

$$f(n,l) = \sum_{k=0}^{n-1}\lfloor l/2^k\rfloor, \quad (B.17)$$

$$S(n,l) = \sum_{k=1}^{n}e^{i(\pi f(k,l)-k\lambda_n)}, \quad (B.18)$$

$$G(l) = 1 + (-1)^l e^{2i\delta_n}, \quad (B.19)$$

$$H(n,l) = \frac{e^{-in\delta_n}}{2^n}\prod_{k=0}^{n-1}G(\lfloor l/2^k\rfloor), \quad (B.20)$$

$$J(n,l) = \sum_{k=0}^{n-1}e^{i(\pi f(k,\lfloor l/2^{n-k}\rfloor)-k\lambda_n)}, \quad (B.21)$$

$$Z(n,l) = \sum_{k=0}^{n-1}S(k,\lfloor l/2^{n-k}\rfloor), \quad (B.22)$$

and reads

$$U_n^n|\alpha\rangle = \sum_{k=0}^{2^n-1}H(n,k)e^{\gamma_n Re(\alpha J(n,k)+\gamma_n Z(n,k))}\left|e^{i(\pi f(n,k)-n\lambda_n)}\alpha + S(n,k)\gamma_n\right\rangle. \quad (B.23)$$

This expression completely determines the time evolution of a coherent state in this model.

Our ultimate goal would be to take the limit $n \to \infty$ of (B.23). To gain insight into how we could achieve this, we shall first examine the amplitudes implied by this expression. Specifically, coherent-to-coherent state amplitudes.

## B.2 Scaling of Trotter discretization and comparison with Hilbert-space truncation

To clarify the computational scaling discussed in the main text, we analyze how the number of Trotter slices $n$ relates to the model parameters, and compare this with the Hilbert-space cutoff $N_{\text{max}}$ required in direct diagonalization.

In the Rabi Hamiltonian,

$$\mathcal{H} = \omega_0 \sigma^z + \omega a^\dagger a + g \sigma^x (a + a^\dagger), \tag{B.24}$$

the coherent states $|\alpha\rangle$ and $|\beta\rangle$ have a Poissonian occupation distribution $p_N = \frac{|\alpha|^{2N} e^{-|\alpha|^2}}{N!}$ in terms of the harmonic oscillator basis (with its basis states labeled by $N$), which extends up to $N \approx |\alpha|^2$. Therefore, the cutoff required for numerical convergence in exact diagonalization scales as $N_{\max} \sim |\alpha|^2$.

In the Trotter decomposition, time evolution is approximated as

$$e^{-it\mathcal{H}} \approx \left( e^{-i\frac{t}{n}\mathcal{H}_0} e^{-i\frac{t}{n}\mathcal{H}_{\text{int}}} \right)^n, \tag{B.25}$$

with a global error that scales as

$$\|e^{-it\mathcal{H}} - U_n^n\| \lesssim \frac{C_1 t^2}{n} \|[\mathcal{H}_0, \mathcal{H}_{\text{int}}]\|, \tag{B.26}$$

where $C_1$ is a numerical constant of order unity (see Refs. [39–42]). For the Rabi Hamiltonian, the dominant commutator behaves as $[\mathcal{H}_0, \mathcal{H}_{\text{int}}] \sim g \omega \sigma^x (a^\dagger - a)$, whose expectation value scales as $\|[\mathcal{H}_0, \mathcal{H}_{\text{int}}]\| \sim g \omega |\alpha|$. Requiring the Trotter error to remain below a tolerance $\varepsilon$ gives

$$n \gtrsim C_1 \frac{g \omega |\alpha| t^2}{\varepsilon}. \tag{B.27}$$

Hence, while $N_{\max}$ quantifies the required Hilbert-space size, $n$ determines the temporal resolution. Their respective dependencies, $N_{\max} \sim |\alpha|^2$ and $n \propto |\alpha|$, illustrate the more favorable scaling of the Trotter-Ising formulation.

In numerical applications, the tolerance $\varepsilon$ is typically chosen between $10^{-5}$ and $10^{-6}$ for imaginary-time simulations (see what follows) and somewhat larger ($10^{-4}$-$10^{-5}$) for real-time evolution, balancing stability and computational cost. Even with these stringent tolerances, the linear scaling of cost with $n$ makes the Ising mapping significantly more efficient than Hilbert-space truncation approaches, which scale as $\mathcal{O}(N_{\max}^2)$-$\mathcal{O}(N_{\max}^3)$.

A completely analogous argument applies for imaginary time evolution. In this case the Trotter approximation is

$$e^{-\tau\mathcal{H}} \approx \left( e^{-\frac{\tau}{n}\mathcal{H}_0} e^{-\frac{\tau}{n}\mathcal{H}_{\text{int}}} \right)^n, \tag{B.28}$$

with a global error that scales as

$$\|e^{-\tau\mathcal{H}} - U_n^n\| \lesssim \frac{C_1 \tau^2}{n} \|[\mathcal{H}_0, \mathcal{H}_{\text{int}}]\|. \tag{B.29}$$

Just like before, requiring the Trotter error to remain below a tolerance $\varepsilon$ gives

$$n \gtrsim C_1 \frac{g \omega |\alpha| \tau^2}{\varepsilon}. \tag{B.30}$$

We note that even though the nature of the deviations in this case — mostly controlled by decaying exponentials — are different than in real time, where oscillatory phases appear, the same global bound applies.

This analysis thus confirms that the present formulation offers a scalable and accurate route for both real- and imaginary-time simulations of the Rabi model and related spin-boson systems.

To understand why the apparent exponential growth in configurations does not spoil this efficiency, we analyze the internal structure of the Trotter-Ising representation.

For a fixed number of slices $n$, the kernels $K_{j,\ell}$ and sources $B_\ell^\pm$ are oscillatory in $(j,\ell)$, except for the local term $\delta_{\ell,j+1}$. The suppression of most spin configurations $\{s_\ell\}$ arises not from local decay but from two mechanisms: (i) a nearest-neighbor domain-wall penalty and (ii) destructive phase interference in the long-range and linear couplings.

The complex action is

$$\mathcal{S}[s] = \frac{1}{2} \sum_{j,\ell=1}^{n-1} s_j s_\ell K_{j,\ell} + \sum_{\ell=1}^{n-1} s_\ell B_\ell^\pm, \qquad s_\ell \in \{-1, 1\}. \tag{B.31}$$

The nearest-neighbor contribution yields

$$J_n^{\mathrm{R}} = \Re[\ln(i \cot \delta_n)] = \ln|\cot \delta_n|, \qquad J_n^{\mathrm{E}} = \ln\left(\coth \frac{P\omega_0 \tau}{n}\right) > 0. \tag{B.32}$$

For practical Trotter choices ($n$ large), $\delta_n \ll 1$ and $J_n^{\mathrm{R}} \simeq \ln n - \ln(P\omega_0 t) > 0$, so both real and imaginary time favor spin alignment. If $m$ is the number of domain walls, the real part of the action scales as $\Re \mathcal{S}_{\mathrm{NN}}[s] \sim m J_n$, exponentially suppressing configurations with many domain walls.

The long-range part of the kernel in real time,

$$K_{j,\ell}^{(\mathrm{LR,R})} = \gamma_n^2 e^{-i|j-\ell|\lambda_n}, \tag{B.33}$$

contributes

$$\sum_{j,\ell} s_j s_\ell e^{-i|j-\ell|\lambda_n} = (n-1) + 2\sum_{d=1}^{n-2} \cos(d\lambda_n) C_d(s), \qquad C_d(s) = \sum_{j=1}^{n-1-d} s_j s_{j+d}. \tag{B.34}$$

Using $|C_d(s)| \le n$ and Dirichlet-kernel bounds,

$$\left|\sum_{j,\ell} s_j s_\ell e^{-i|j-\ell|\lambda_n}\right| \lesssim n + \frac{n}{|\sin(\lambda_n/2)|}, \tag{B.35}$$

yielding

$$\Re\left[\frac{1}{2}\gamma_n^2 \sum_{j,\ell} s_j s_\ell e^{-i|j-\ell|\lambda_n}\right] = \mathcal{O}\left(\frac{g^2 t}{\omega}\right), \tag{B.36}$$

which is finite and independent of $n$. In imaginary time, since $K_{j,\ell}^{(\mathrm{LR,E})} > 0$, the bound improves to $\mathcal{O}((gt)^2/n)$, vanishing as $n \to \infty$.

The linear source term,

$$B_\ell^\pm = \gamma_n \left[\beta^* e^{-i\ell\lambda_n} \pm \alpha e^{-i(n-\ell)\lambda_n}\right], \tag{B.37}$$

obeys

$$\left|\sum_\ell s_\ell B_\ell^\pm\right| \le (gt)(|\alpha| + |\beta|), \tag{B.38}$$

independent of $n$, and tighter in imaginary time due to exponential decay.

Combining all contributions,

$$|e^{\mathcal{S}[s]}| \le \exp\left[m J_n + C_1 \frac{g^2 t}{\omega} + C_2(gt)(|\alpha| + |\beta|)\right], \tag{B.39}$$

where $C_1$ and $C_2$ are positive constants, independent of $n$ and of order unity ($\mathcal{O}(1)$). Thus, for fixed $n$, configurations with many domain walls are exponentially suppressed, while the nonlocal and linear terms remain bounded.

The oscillatory structure selects spin domains of typical size $\Delta i \sim n/(\omega t)$, implying a characteristic number of domain walls $m_\star \sim \omega t$, independent of $n$. Large $|\alpha|, |\beta|$ values further stabilize these domains without changing this scaling.

Grouping configurations by $m$,

$$\sum_{m>M} \sum_{s:\#\mathrm{DW}(s)=m} |e^{\mathcal{S}[s]}| \le e^{C_1 \frac{g^2 t}{\omega} + C_2(gt)(|\alpha|+|\beta|)} \sum_{m>M} \binom{n}{m} e^{mJ_n}. \tag{B.40}$$

For $J_n > 0$, the tail becomes negligible for $M \gtrsim m_\star \sim \omega t$, independent of $n$, implying polynomial computational scaling in $n$ (and linear in $M$).

In the $\omega_0$-expansion,

$$\left| \sum_{m>M} \frac{(iP\omega_0 t)^m}{m!} F_m \right| \le e^{C_3(gt)(|\alpha|+|\beta|)+C_4 \frac{g^2 t}{\omega}} \sum_{m>M} \frac{(|P\omega_0 t|)^m}{m!} \Xi_m, \tag{B.41}$$

where $\Xi_m$ bounds $|F_m|$, $C_3$ and $C_4$ are positive constants, independent of $n$ and of order unity ($\mathcal{O}(1)$). Factorial decay ensures convergence, and choosing $M \sim \omega t$ captures the dominant contribution with tolerance independent of $n$.

In summary, even though the formal sum involves $2^n$ configurations, only $\mathcal{O}(\omega t)$ of them contribute significantly due to domain-wall suppression and bounded long-range interactions. Combined with the global Trotter-error bound, this shows that the Trotter-Ising formulation achieves controlled accuracy and polynomial computational scaling for both real- and imaginary-time evolutions.

# C  Derivation of dot product formula

The action of one Trotterized unitary time evolution operator gave us:

$$U_n |\alpha\rangle = e^{\gamma_n Re(\alpha)} \Big[ \cos \delta_n \left| (\alpha + \gamma_n) e^{-i\lambda_n} \right\rangle - i \sin \delta_n \left| -(\alpha + \gamma_n) e^{-i\lambda_n} \right\rangle \Big]. \tag{C.1}$$

Applying again, we obtain for $U_n^2$

$$U_n^2 |\alpha\rangle = e^{\gamma_n Re(\alpha)} \Bigg\{ \cos \delta_n e^{\gamma_n Re((\alpha+\gamma_n)e^{-i\lambda_n})} \Big[ \cos \delta_n \left| ((\alpha+\gamma_n)e^{-i\lambda_n}+\gamma_n)e^{-i\lambda_n} \right\rangle \tag{C.2}$$
$$- i \sin \delta_n \left| -((\alpha+\gamma_n)e^{-i\lambda_n}+\gamma_n)e^{-i\lambda_n} \right\rangle \Big]$$
$$- i \sin \delta_n e^{\gamma_n Re((\alpha+\gamma_n)e^{-i\lambda_n})} \Big[ \cos \delta_n \left| (-(\alpha+\gamma_n)e^{-i\lambda_n}+\gamma_n)e^{-i\lambda_n} \right\rangle$$
$$- i \sin \delta_n \left| -(-(\alpha+\gamma_n)e^{-i\lambda_n}+\gamma_n)e^{-i\lambda_n} \right\rangle \Big] \Bigg\},$$

and for $U_n^3$

$$U_n^3 |\alpha\rangle = e^{\gamma_n Re(\alpha)} \Bigg\{ \cos \delta_n e^{\gamma_n Re(\beta_1)} \Big[ \cos \delta_n e^{\gamma_n Re(\beta_2)} \Big( \cos \delta_n \left| (\beta_2 + \gamma_n) e^{-i\lambda_n} \right\rangle \tag{C.3}$$
$$- i \sin \delta_n \left| -(\beta_2 + \gamma_n) e^{-i\lambda_n} \right\rangle \Big)$$
$$- i \sin \delta_n e^{-\gamma_n Re(\beta_2)} \Big( \cos \delta_n \left| (-\beta_2 + \gamma_n) e^{-i\lambda_n} \right\rangle$$
$$- i \sin \delta_n \left| -(-\beta_2 + \gamma_n) e^{-i\lambda_n} \right\rangle \Big) \Big]$$

$$-i\sin\delta_n e^{-\gamma_n Re(\beta_1)}\Big[\cos\delta_n e^{-\gamma_n Re(\beta_2)}\Big(\cos\delta_n \big|(-\beta_2+\gamma_n)e^{-i\lambda_n}\big\rangle$$
$$-i\sin\delta_n \big|-(-\beta_2+\gamma_n)e^{-i\lambda_n}\big\rangle\Big)$$
$$-i\sin\delta_n e^{\gamma_n Re(\beta_2)}\Big(\cos\delta_n \big|(\beta_2+\gamma_n)e^{-i\lambda_n}\big\rangle$$
$$-i\sin\delta_n \big|-(\beta_2+\gamma_n)e^{-i\lambda_n}\big\rangle\Big)\Big]\Big\},$$

where in the last equation we have defined $\beta_1 = (\alpha+\gamma_n)e^{-i\lambda_n}$ and $\beta_2 = (\beta_1+\gamma_n)e^{-i\lambda_n}$.

We'd like a general formula for the term $U_n^n$. Let's rewrite the previous expressions for $n = 0, 1$ and $2$ in a more suggestive way

$$U_n^0|\alpha\rangle = 1 \times |\alpha\rangle,\tag{C.4}$$

$$U_n^1|\alpha\rangle = \begin{pmatrix}\cos\delta_n e^{\gamma_n Re(\alpha)}\\-i\sin\delta_n e^{\gamma_n Re(\alpha)}\end{pmatrix}\begin{pmatrix}\big|(\alpha+\gamma_n)e^{-i\lambda_n}\big\rangle\\\big|-(\alpha+\gamma_n)e^{-i\lambda_n}\big\rangle\end{pmatrix},\tag{C.5}$$

$$U_n^2|\alpha\rangle = \begin{pmatrix}\cos^2\delta_n e^{\gamma_n(Re(\alpha)+Re((\alpha+\gamma_n)e^{-i\lambda_n}))}\\-i\sin\delta_n\cos\delta_n e^{\gamma_n(Re(\alpha)+Re((\alpha+\gamma_n)e^{-i\lambda_n}))}\\-i\sin\delta_n\cos\delta_n e^{\gamma_n(Re(\alpha)+Re(-(\alpha+\gamma_n)e^{-i\lambda_n}))}\\-\sin^2\delta_n e^{\gamma_n(Re(\alpha)+Re(-(\alpha+\gamma_n)e^{-i\lambda_n}))}\end{pmatrix}\cdot\begin{pmatrix}\big|((\alpha+\gamma_n)e^{-i\lambda_n}+\gamma_n)e^{-i\lambda_n}\big\rangle\\\big|-((\alpha+\gamma_n)e^{-i\lambda_n}+\gamma_n)e^{-i\lambda_n}\big\rangle\\\big|(-(\alpha+\gamma_n)e^{-i\lambda_n}+\gamma_n)e^{-i\lambda_n}\big\rangle\\\big|-(-(\alpha+\gamma_n)e^{-i\lambda_n}+\gamma_n)e^{-i\lambda_n}\big\rangle\end{pmatrix}.\tag{C.6}$$

We generalize the previous result for abirtrary $n$ such that we have the multiplication of two vectors of length $2^n$:

$$U_n^n|\alpha\rangle = \begin{pmatrix}b_n^0\\b_n^1\\\vdots\\b_n^{2^n-1}\end{pmatrix}\begin{pmatrix}\big|a_n^0\big\rangle\\\big|a_n^1\big\rangle\\\vdots\\\big|a_n^{2^n-1}\big\rangle\end{pmatrix},\tag{C.7}$$

where the coefficients are given by a branching recurrence relation given by the base case $b_0^0 = 1$ and $a_0^0 = \alpha$. The recurrences are

$$a_n^{2k-1} = e^{-i\lambda_n}\left(a_{n-1}^{k-1}+\gamma_n\right),\tag{C.8}$$

$$a_n^{2k} = -e^{-i\lambda_n}\left(a_{n-1}^k+\gamma_n\right),\tag{C.9}$$

$$b_n^{2k-1} = \cos\delta_n\, b_{n-1}^{k-1}e^{\gamma_n Re(a_{n-1}^{k-1})},\tag{C.10}$$

$$b_n^{2k} = -i\sin\delta_n\, b_{n-1}^k e^{\gamma_n Re(a_{n-1}^k)}.\tag{C.11}$$

## C.1 Recurrence in the $a$ vector components

We can rewrite the recurrence in a simpler way to avoid the branching depending on the vector component by using the floor function. Therefore,

$$a_n^l = e^{i\pi l}e^{-i\lambda_n}\left(a_{n-1}^{\lfloor l/2\rfloor}+\gamma_n\right).\tag{C.12}$$

The base case is $a_0^0 = \alpha$. By simple inspection we see that

$$a_1^l = e^{i(\pi\lfloor l\rfloor-\lambda_n)}\left(a_0^{\lfloor l/2\rfloor}+\gamma_n\right) = e^{i(\pi\lfloor l\rfloor-\lambda_n)}\alpha + e^{i(\pi\lfloor l\rfloor-\lambda_n)}\gamma_n,\tag{C.13}$$

$$a_2^l = e^{i(\pi\lfloor l\rfloor-\lambda_n)}\left(a_1^{\lfloor l/2\rfloor}+\gamma_n\right) = e^{i(\pi[\lfloor l\rfloor+\lfloor l/2\rfloor]-2\lambda_n)}\alpha + e^{-i\lambda_n}\left(e^{i\pi\lfloor l\rfloor}+e^{i\pi\lfloor l/2\rfloor}\right)\gamma_n,\tag{C.14}$$

$$a_3^l = e^{i(\pi\lfloor l\rfloor-\lambda_n)}\left(a_2^{\lfloor l/2\rfloor}+\gamma_n\right) = e^{i(\pi[\lfloor l\rfloor+\lfloor l/2\rfloor+\lfloor l/4\rfloor]-3\lambda_n)}\alpha + e^{-i\lambda_n}\left(e^{i\pi\lfloor l\rfloor}+e^{i\pi\lfloor l/2\rfloor}+e^{i\pi\lfloor l/4\rfloor}\right)\gamma_n.\tag{C.15}$$

Solving this recurrence is nontrivial since it involves a recurrence in two indexes, and it also has non-constant coefficients. Luckily, after many trials, we found a solution by simple inspection:

$$a_n^l = e^{i(\pi f(n,l) - n\lambda_n)}\alpha + S(n,l)\gamma_n \,, \tag{C.16}$$

where

$$f(n,l) \equiv \sum_{k=0}^{n-1} \lfloor l/2^k \rfloor \,, \tag{C.17}$$

$$S(n,l) \equiv \sum_{k=1}^{n} e^{i(\pi f(k,l) - k\lambda_n)} \,. \tag{C.18}$$

## C.2 Recurrence in the $b$ vector components

Applying the same logic we used for the previous recurrence, we can write the recurrence equation for $b_n^l$ in a more compact way using the floor function:

$$b_n^l = \frac{e^{-i\delta_n}}{2}(1 + (-1)^l e^{2i\delta_n})b_{n-1}^{\lfloor l/2 \rfloor}e^{\gamma_n Re(a_{n-1}^{\lfloor l/2 \rfloor})} = \frac{e^{-i\delta_n}}{2}G(l)b_{n-1}^{\lfloor l/2 \rfloor}e^{\gamma_n Re(a_{n-1}^{\lfloor l/2 \rfloor})} \,, \tag{C.19}$$

where $G(l) = 1 + (-1)^l e^{2i\delta_n}$. The base case is $b_0^0 = 1$. By simple inspection we see that

$$b_1^l = \frac{e^{-i\delta_n}}{2}G(l)b_0^0 e^{\gamma_n Re(a_0^{\lfloor l/2 \rfloor})} = \frac{e^{-i\delta_n}}{2}G(l)e^{\gamma_n Re(a_0^{\lfloor l/2 \rfloor})} \,, \tag{C.20}$$

$$b_2^l = \frac{e^{-i\delta_n}}{2}G(l)b_1^{\lfloor l/2 \rfloor}e^{\gamma_n Re(a_1^{\lfloor l/2 \rfloor})} = \frac{e^{-2i\delta_n}}{4}G(\lfloor l \rfloor)G(\lfloor l/2 \rfloor)e^{\gamma_n Re(a_0^{\lfloor l/4 \rfloor} + a_1^{\lfloor l/2 \rfloor})} \,, \tag{C.21}$$

$$b_3^l = \frac{e^{-i\delta_n}}{2}G(l)b_2^{\lfloor l/2 \rfloor}e^{\gamma_n Re(a_2^{\lfloor l/2 \rfloor})} = \frac{e^{-3i\delta_n}}{8}G(\lfloor l \rfloor)G(\lfloor l/2 \rfloor)G(\lfloor l/4 \rfloor)e^{\gamma_n Re(a_0^{\lfloor l/8 \rfloor} + a_1^{\lfloor l/4 \rfloor} + a_2^{\lfloor l/2 \rfloor})} \,. \tag{C.22}$$

Let's find a solution to the recurrence. Defining the following function (for which we have not found a closed form):

$$H(n,l) \equiv \frac{e^{-in\delta_n}}{2^n}\prod_{k=0}^{n-1}G(\lfloor l/2^k \rfloor) \,, \tag{C.23}$$

we have

$$b_n^l = H(n,l)\exp\left(\gamma_n Re\Big[\sum_{k=0}^{n-1}a_k^{\lfloor l/2^{n-k} \rfloor}\Big]\right) \,. \tag{C.24}$$

Let's calculate the term inside the exponential with more detail:

$$\sum_{k=0}^{n-1}a_k^{\lfloor l/2^{n-k} \rfloor} = \alpha\left(\sum_{k=0}^{n-1}e^{i(\pi f(k,\lfloor l/2^{n-k} \rfloor) - k\lambda_n)}\right) + \gamma_n\left(\sum_{k=0}^{n-1}S\big(k,\lfloor l/2^{n-k} \rfloor\big)\right) \,. \tag{C.25}$$

Furthermore, let's note that:

$$f(k,\lfloor l/2^{n-k} \rfloor) = \sum_{j=0}^{k-1}\lfloor l/2^{n-k+j} \rfloor \equiv r(k,l) \,, \tag{C.26}$$

and

$$S(k,\lfloor l/2^{n-k} \rfloor) = \sum_{j=1}^{k}e^{i(\pi \sum_{i=0}^{j-1}\lfloor l/2^{n-k+i} \rfloor - j\lambda_n)} \equiv T(k,l) \,. \tag{C.27}$$

Then we have

$$\sum_{k=0}^{n-1} a_k^{\lfloor l/2^{n-k} \rfloor} = \alpha \left( \sum_{k=0}^{n-1} e^{i(\pi r(k,l) - k\lambda_n)} \right) + \gamma_n \sum_{k=0}^{n-1} T(k,l) . \tag{C.28}$$

Finally, we can define

$$J(n,l) \equiv \sum_{k=0}^{n-1} e^{i(\pi r(k,l) - k\lambda_n)} , \tag{C.29}$$

$$Z(n,l) \equiv \sum_{k=0}^{n-1} T(k,l) , \tag{C.30}$$

with which we have

$$\sum_{k=0}^{n-1} a_k^{\lfloor l/2^{n-k} \rfloor} = \alpha J(n,l) + \gamma_n Z(n,l) , \tag{C.31}$$

and we therefore conclude that the solution to the recurrence equation for $b_n^l$ is:

$$b_n^l = H(n,l) e^{\gamma_n Re(\alpha J(n,l) + \gamma_n Z(n,l))} . \tag{C.32}$$

# D  Coherent to coherent state amplitudes

One of the noteworthy features of (B.23) is that all of the dependence on the sum index $k$ in the functions $H, J, Z, S$ appears through the features of its dyadic expansion. That is, $k$ is used as an integer number generator via quotients with $2^j$. Moreover, it only goes into the result through the value of $e^{i\pi \lfloor l/2^j \rfloor}$, which is always $\pm 1$. That means we are using integer numbers to build binary numbers, or equivalently, sign sequences

$$k \in \mathbb{Z} \text{ s.t. } k < 2^n \quad \Longleftrightarrow \quad \{s_\ell\}_{\ell=1}^n(k) = \{s_\ell | s_\ell = e^{i\pi \lfloor k/2^\ell \rfloor}\} . \tag{D.1}$$

Therefore, the sum can be rewritten as a sum over sign choices. Taking the floor function of integers divided by powers of 2 is one way to do this, but any way of generating all possible sign sequences of size $n$ is an equivalent description.

With the correspondence we just outlined, we can write the above functions as

$$S(n,k) = \sum_{\ell=1}^n \left( \prod_{j=0}^{\ell-1} s_j(k) \right) e^{-i\ell\lambda_n} , \tag{D.2}$$

$$H(n,k) = \frac{(-i\sin(\delta_n)\cos(\delta_n))^{n/2}}{(-i\tan(\delta_n))^{\frac{1}{2}\sum_{j=0}^{n-1} s_j(k)}} , \tag{D.3}$$

$$J(n,k) = \sum_{\ell=0}^{n-1} \left( \prod_{j=0}^{\ell-1} s_{n-\ell+j}(k) \right) e^{-i\ell\lambda_n} , \tag{D.4}$$

$$Z(n,k) = \sum_{\ell=0}^{n-1} \sum_{j=0}^{\ell-1} \left( \prod_{i=0}^{j-1} s_{n-\ell+i}(k) \right) e^{-ij\lambda_n} , \tag{D.5}$$

where we have used that

$$\text{\# of positive signs} = \frac{n + \sum_{j=0}^{n-1} s_j(k)}{2} , \tag{D.6}$$

$$\text{\# of negative signs} = \frac{n - \sum_{j=0}^{n-1} s_j(k)}{2} . \tag{D.7}$$

Now we may drop the $k$ label altogether and write

$$S_n[s] = \sum_{\ell=1}^{n} \left( \prod_{j=0}^{\ell-1} s_j \right) e^{-i\ell\lambda_n} , \tag{D.8}$$

$$H_n[s] = \frac{(-i\sin(\delta_n)\cos(\delta_n))^{n/2}}{(-i\tan(\delta_n))^{\frac{1}{2}\sum_{j=0}^{n-1} s_j}} , \tag{D.9}$$

$$J_n[s] = \sum_{\ell=0}^{n-1} \left( \prod_{j=0}^{\ell-1} s_{n-\ell+j} \right) e^{-i\ell\lambda_n} , \tag{D.10}$$

$$Z_n[s] = \sum_{\ell=0}^{n-1}\sum_{j=0}^{\ell-1} \left( \prod_{i=0}^{j-1} s_{n-\ell+i} \right) e^{-ij\lambda_n} , \tag{D.11}$$

so that a general transition amplitude can be written as

$$\langle\beta|U_n^n|\alpha\rangle = \sum_{\{s_k\}_{k=0}^{n-1}} H_n[s]\exp\left(\gamma_n\mathrm{Re}\{\alpha J_n[s] + \gamma_n Z_n[s]\}\right)$$
$$\times \langle\beta|e^{-i\omega t}\left(\prod_{k=0}^{n-1} s_k\right)\alpha + S_n[s]\gamma_n\rangle . \tag{D.12}$$

Furthermore, since $\langle\beta|\alpha\rangle = \exp(-[|\alpha|^2 + |\beta|^2 - 2\beta^*\alpha]/2)$, and using that

$$S_n[s] = e^{-i\omega t}\left(\prod_{j=0}^{n-1} s_j\right) J_n^*[s] ,$$

we can get a more explicit expression for the transition amplitude:

$$\langle\beta|U_n^n|\alpha\rangle = \sum_{\{s_k\}_{k=0}^{n-1}} H_n[s]\exp\left(\gamma_n^2 Z_n[s] + \gamma_n[\beta^* S_n[s] + \alpha J_n[s]]\right)$$
$$\times \exp\left(-\frac{|\alpha|^2 - 2e^{-i\omega t}\left(\prod_{j=0}^{\ell-1} s_j\right)\beta^*\alpha + |\beta|^2}{2}\right) . \tag{D.13}$$

One can now rearrange the sum in terms of the accumulated sign variable

$$s_\ell = \prod_{j=0}^{\ell-1} s_j , \tag{D.14}$$

with $s_0 = 1$. In this representation, the defining functions become

$$S_n[s] = \sum_{\ell=1}^{n} s_\ell e^{-i\ell\lambda_n} , \tag{D.15}$$

$$H_n[s] = \left(\frac{-i\sin(2\delta_n)}{2}\right)^{n/2}\exp\left(\frac{\ln(i\cot(\delta_n))}{2}\sum_{j=0}^{n-1} s_j s_{j+1}\right) , \tag{D.16}$$

$$J_n[s] = S_n^*[s] s_n e^{-i\omega t} , \tag{D.17}$$

$$Z_n[s] = \sum_{\ell=1}^{n-1}\sum_{j=1}^{\ell} s_{n-\ell+j} s_{n-\ell} e^{-ij\lambda_n} , \tag{D.18}$$

which are all at most quadratic in $s_j$. This means we can interpret this Trotterized time evolution as an Ising model with complex long-range interactions in presence of an inhomogeneous magnetic field. To see this explicitly, first note that we can write

$$\gamma_n^2 Z_n[s] \underset{n\to\infty}{=} \gamma_n^2 \frac{1}{2} \sum_{\ell,j=1}^{n-1} s_j s_\ell e^{-i|j-\ell|\lambda_n}, \tag{D.19}$$

where the equality holds only in the limit $n \to \infty$, i.e., up to terms that contribute multiplicatively to the full amplitude as $e^{1/n}$, and therefore can be neglected. Similarly, we also have

$$\gamma_n[\beta^* S_n[s] + \alpha J_n[s]] \underset{n\to\infty}{=} \gamma_n \sum_{\ell=1}^{n-1} s_\ell \left[ \beta^* e^{-i\ell\lambda_n} + \alpha s_n e^{-i(n-\ell)\lambda_n} \right]. \tag{D.20}$$

Put together, these lead to the sum of two expressions alike the partition function of an Ising model with a particular source term, which is different for each spin variable $s_n$. Then, we have (Eq. (17) in the main text),

$$\langle \beta | U_n^n | \alpha \rangle \underset{n\to\infty}{=} \left( \frac{-i\sin(2\delta_n)}{2} \right)^{n/2} e^{-\frac{|\alpha|^2+|\beta|^2}{2}} \tag{D.21}$$

$$\times \left\{ \sum_{\substack{\{s_k\}_{k=1}^{n-1} \\ s_0=s_n=1}} e^{\frac{1}{2}\sum_{j,\ell=1}^{n-1} s_j s_\ell K_{j,\ell} + \sum_{\ell=1}^{n-1} s_\ell B_\ell^+ + e^{-i\omega t}\beta^*\alpha} + \sum_{\substack{\{s_k\}_{k=1}^{n-1} \\ s_0=-s_n=1}} e^{\frac{1}{2}\sum_{j,\ell=1}^{n-1} s_j s_\ell K_{j,\ell} + \sum_{\ell=1}^{n-1} s_\ell B_\ell^- - e^{-i\omega t}\beta^*\alpha} \right\},$$

where

$$K_{j,\ell} = \delta_{\ell,j+1} \ln(i\cot\delta_n) + \gamma_n^2 e^{-i|j-\ell|\lambda_n}, \tag{D.22}$$

$$B_\ell^\pm = \gamma_n \left[ \beta^* e^{-i\ell\lambda_n} \pm \alpha e^{-i(n-\ell)\lambda_n} \right]. \tag{D.23}$$

These expressions have exactly the form of an Ising model, and their continuum limit is well-defined for all pieces that involve $\gamma_n$ as a factor, because the sums over $j$ and $\ell$ can be directly expressed in terms of integrals that involve a continuous version of the sign variables in the sum $s_\ell \to s(x)$. When there is a finite number of sign flips, the conversion can be made directly. In more general cases, one must interpret $s(x)$ as a local average of the "microscopic" $s_\ell$ variables.

The more difficult part to handle, where the continuum limit is not trivially obtained, is the factor that involves $\delta_n$.

One way to rearrange the sum in (D.21) is by counting the number of changes of sign in the sequence $\{s_k\}_{k=0}^n$. Let's say that there are $m$ sign flips in a given sequence $s$. Then, one has

$$H_n[s] = (\cos\delta_n)^{n/2}(-i\tan\delta_n)^m, \tag{D.24}$$

which in the limit $n \to \infty$, at fixed $\omega_0 t$, becomes simply $(-i\omega_0 t/n)^m$. The $(1/n)^m$ factor serves the purpose of averaging over all the possible positions where the sign flip in the sequence $s$ was inserted. One can then take the limit $n \to \infty$ and obtain

$$\langle \beta | U(t) | \alpha \rangle = e^{-\frac{|\alpha|^2+|\beta|^2}{2}} \sum_{m=0}^{\infty} \frac{(iP\omega_0 t)^m}{m!} \left\langle e^{-g^2 t^2 Z[s] - igt\left[\alpha J[s] + \beta^* s(1)e^{-i\omega t}J^*[s]\right] + \alpha\beta^* s(1)e^{-i\omega t}} \right\rangle_m, \tag{D.25}$$

where the expectation value $\langle \cdot \rangle_m$ represents an average over all possible configurations of $s(x) \in \{-1,1\}$ such that there are $m$ sign flips. Explicitly, at each fixed $m$, the average is defined as

$$\langle G \rangle_m = m! \int_0^1 dz_m \int_0^{z_m} dz_{m-1} \ldots \int_0^{z_2} dz_1 \, G(z_1, z_2, \ldots, z_m), \tag{D.26}$$

where $\{z_i\}_{i=1}^m$ are the positions at which $s(x)$ flips sign. The functionals $Z[s]$ and $J[s]$ are the continuum counterparts of $Z_n[s]$ and $J_n[s]$:

$$Z[s] = \frac{1}{2} \int_0^1 dx \int_0^1 dy \, s(x)s(y) e^{-i\omega t |x-y|}, \tag{D.27}$$

$$J[s] = s(1)e^{-i\omega t} \int_0^1 dx \, s(x) e^{i\omega t x}. \tag{D.28}$$

Assuming we have an ordered list of sign flip positions $\{z_i\}_{i=1}^m$, we can write expressions for $Z$ and $J$ at each value of $m$ directly in terms of the sign flip positions:

$$-g^2 t^2 Z[s] = -\frac{4g^2}{\omega^2} \sum_{k=1}^m \sum_{\ell=1}^{k-1} (-1)^{k+\ell} e^{-i(z_k - z_\ell)\omega t} - \frac{2g^2}{\omega^2} \sum_{k=1}^m (-1)^k \left[ e^{-iz_k \omega t} - (-1)^m e^{-i(1-z_k)\omega t} \right]$$

$$- \frac{g^2}{\omega^2} \left( 1 - (-1)^m e^{-i\omega t} \right) - \frac{2mg^2}{\omega^2} + \frac{ig^2 t}{\omega}, \tag{D.29}$$

$$igt J[s] = \frac{g}{\omega} e^{-i\omega t} (-1)^m \sum_{k=0}^m (-1)^k \left( e^{iz_{k+1}\omega t} - e^{iz_k \omega t} \right). \tag{D.30}$$

With this, it is convenient to introduce the functions

$$F_m = \left\langle e^{-g^2 t^2 Z[s] - igt \left[ \alpha J[s] + \beta^* s(1) e^{-i\omega t} J^*[s] \right] - \alpha\beta^* \left[ 1 - s(1)e^{-i\omega t} \right]} \right\rangle_m e^{\frac{2mg^2}{\omega^2} + \left(\alpha + \frac{g}{\omega}\right)\left(\beta^* + \frac{g}{\omega}\right)\left[1 - (-1)^m e^{-i\omega t}\right] - \frac{ig^2 t}{\omega}}. \tag{D.31}$$

We extract the prefactor in the second line to isolate the dependence on the sign flips into a single expression. That is to say, we have defined $F_m$ such that $F_0 = 1$, and factorized out all terms that do not depend on the sign flip positions for $m > 0$. To be explicit, in terms of $F_m$ the amplitude reads

$$\langle \beta | U(t) | \alpha \rangle = e^{-\frac{|\alpha|^2 - 2\beta^* \alpha + |\beta|^2}{2}} e^{\frac{ig^2 t}{\omega}} \sum_{m=0}^\infty \frac{(iP\omega_0 t)^m}{m!} e^{-\frac{2mg^2}{\omega^2} - \left(\alpha + \frac{g}{\omega}\right)\left(\beta^* + \frac{g}{\omega}\right)\left[1 - (-1)^m e^{-i\omega t}\right]} F_m, \tag{D.32}$$

and $F_m$ is given by

$$F_m = \left\langle \exp\left( -\frac{4g^2}{\omega^2} \sum_{k=1}^m \sum_{\ell=1}^{k-1} (-1)^{k+\ell} e^{-i(z_k - z_\ell)\omega t} - \frac{2g}{\omega} \sum_{k=1}^m \left[ \left(\beta^* + \frac{g}{\omega}\right)(-1)^k e^{-iz_k \omega t} \right.\right.\right. \tag{D.33}$$

$$\left.\left.\left. - \left(\alpha + \frac{g}{\omega}\right)(-1)^{m+k} e^{-i(1-z_k)\omega t} \right] \right) \right\rangle_m,$$

where the average is taken over the possible sign flip positions $z_k \in (0,1)$, with $\{z_k\}_{k=1}^m$ an ordered sequence.

Equation (D.32), (which corresponds to Eq.(17) in the main text), is a well-defined power series in $\omega_0$, of which one can numerically compute each term.

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
