# Peer review of "A correspondence between the Rabi model and an Ising model with long-range interactions"

_SciPost Physics, doi:SciPost Phys. 19, 153 (2025)_

## Round 1 · Referee Report · Anonymous (Referee 1) · 2025-8-29

Report
The article under review establishes an interesting connection between the Rabi model (an important paradigm of quantum optics) and classical thermodynamics, by expressing the time-dependent transition amplitudes in terms of a partition function of a long-range classical Ising model.
I enjoyed reading the paper, which is well-written (except for a confusing point on the definition of coherent states - see below) and describes nontrivial and beautiful mathematical manipulations, through which the Authors work out in detail the relation between the two formulations. The main shortcoming of the article is, I would say, that the connection does not reveal new features of the two systems. After all, the Rabi model only involves a single atom in a cavity and its properties are relatively straightforward to extract in a direct manner.
This said, the Authors point out in Section 4 that the thermodynamic formulation of the transition amplitudes could be advantageous when the coherent states have large amplitudes, as it allows to avoid the use of a large Hilbert spaces (see After Eq. 22). A similar comment applies to the large-temperature limit of the partition function in Eq. 26 (see after Eq. 28). Therefore, the new formulation of the time evolution already offers some technical advantages. Furthermore, the method could have more general applicability. This is noted in the conclusion, where the Authors propose that the application of a similar approach to other quantum model could be an interesting direction for future studies.
I agree that the idea of this paper could have general applicability. In fact, it basically coincides with the well known quantum-to-classical correspondence in statistical mechanics, where a quantum model with d dimension is mapped to a classical model with d+1 dimensions. Here the Trotterization is performed in real time instead of imaginary time (as we are dealing with a transition amplitude, instead of a partition function). My impression is that this point should be emphasized more prominently in the article, in the introduction or even in the abstract. In the current version, I only noticed such a remark at the end of Section 2.
Finally, the Authors might be interested to consider the limit w/w0 --> 0, when a phase transition to a superradiant phase takes place place at gc=\sqrt{w w0}/2. This topic recently attracted considerable attention, especially following M.-J. Hwang, R. Puebla, and M. B. Plenio, Phys. Rev. Lett. 115, 180404 (2015). An earlier and quite transparent discussion can be also be found in L. Bakemeier, A. Alvermann, and H. Fehske, Phys. Rev. A 85, 043821 (2012). Perhaps the thermodynamic formulation in terms of the Ising model could allow to interpret this phase transition from a different angle. Upon inspection, I do not see this physics emerging in a very obvious way from the classical partition function, therefore I leave this as an optional suggestion.
Requested changes
1) When reading the article, I was confused by the definition of coherent states and amplitudes, in particular Eq. 2 and Eq. 3. For Eq. 2, it is clear that |alpha> is the regular coherent state, satisfying the usual condition a|alpha>= alpha|alpha> (as stated before Eq. 2). However, in this case A_\beta\alpha of Eq. 2 would be a spin operator. Instead, a few lines below A_\beta\alpha is called "amplitude" and amplitudes are usually c-numbers. Even more confusing was for me the following section 2, as |alpha> now indicates a different type of coherent state. The Authors actually note at the end of Section 1 that they use the coherent states of b (and not of a) but I just assumed that they take them in the simple form |alpha>|\pm>, where the spin states are eigenstates of sigma_x. This choice, I would say, is the most intuitive one since b=a sigma_x. However, also in Eq. 3 the spin degrees of freedom do not appear explicitly.
As it turns out, the form of the coherent states is different from my initial guess, and is only given in Appendices A and B. To avoid misunderstandings, I think it is better to define the coherent states of b explicitly in the main text, before introducing the amplitudes. Then, one can appreciate immediately that they are entangled states between spin and cavity with an additional subscript \pm (distinguishing |alpha>_\pm from the usual |alpha>), which is later omitted.
2) I was confused by the meaning of \delta_l,j+1 in Eq. 15 (perhaps because of the concomitant presence of \delta_n). Maybe it is not completely useless to remind the reader that \delta_l,j+1 is a Kronecker delta.
3) Equation A.1 does not coincide with Eq. 1, since \sigma^+\sigma^- is not equal to \sigma_z. The relation on line 284 is wrong and should be corrected. Perhaps it is better to directly use sigma_z in Eq. A.1, like in Eq. 1.
4) I noticed these minor misprints: - there is a space missing on line 283, before "hbar=1" - on line 325, evolution-->evolve? -|0> is missing in on the rightmost side of Eq. B.4
Recommendation
Publish (easily meets expectations and criteria for this Journal; among top 50%)
We appreciate the referee’s overall positive evaluation and the suggestion to emphasize more clearly that the Trotterization in our analysis is carried out in real time. In the revised version that we intend to resubmit, we plan to add sentences in the abstract and introduction highlighting this fact, as well as a further sentence in the introduction to contrast it with the Trotterization in imaginary time that we discuss later on in the manuscript. We also thank the referee for pointing out to us the limit $\omega/\omega_0 \to 0$ and the associated transition to a superradiant phase. While a full analysis of this regime is beyond the present scope of our work, we will mention this in the conclusion as a possible future direction.
Regarding the requested changes: 1) After re-reading our manuscript, we agree with the referee that we should have introduced the (generalized) coherent states in full detail in the introduction, to avoid confusion. We plan to add a couple of paragraphs discussing the differences of these states with the coherent states $|\alpha,\pm\rangle$, and also how one can write one kind of coherent states as a linear combination of the other kind. 2) We will add a sentence doing so. 3) We thank the referee for pointing out this mistake. Indeed, $\sigma^+\sigma^-$ is not equal to $\sigma_z$. We will correct Eq. (A1) by explicitly writing it in terms of $\sigma_z$, in agreement with Eq. (1). In addition to that, we will modify the corresponding sentence on line 284 to be consistent with this correction. 4) We thank the referee for carefully pointing out these misprints. We have corrected all three.

Author: Bruno Scheihing-Hitschfeld on 2025-09-24 [id 5857]
(in reply to Report 2 on 2025-09-03)We thank the referee for the positive assessment under “Strengths” and for pointing out that the analysis of the resulting Ising model was somewhat superficial. We will address this by expanding the discussion of the Ising couplings and their relation to different dynamical regimes of the Rabi model.
Regarding the requested changes: 1) We agree with the referee that we should have introduced the (generalized) coherent states in full detail in the introduction, to avoid confusion. We plan to add a couple of paragraphs discussing the differences of these states with the coherent states $|\alpha,\pm\rangle$, and also how one can write one kind of coherent states as linear combinations of the other kind. 2) We plan to expand the discussion in Section 4 (between Eqs. (23) and (24)), explaining in more detail the form of the Ising couplings $K_{j,\ell}^E$ and how they reflect the different dynamical regimes of the Rabi model. Concretely, we will explain that the kernel $K_{j,\ell}^E$ has two contributions: (i) a nearest-neighbor term, which plays the role of a strong coupling between consecutive time slices in the Trotter decomposition and depends on $\omega_0$, and (ii) a ferromagnetic, exponentially decaying tail whose amplitude grows with $g^2$ and whose range is set by $\omega$. In the continuum limit, this second contribution becomes a nonlocal interaction kernel proportional to $e^{-\omega\tau |u-v|}$. Taking the Fourier transform of this long-range part alone shows that its spectral weight has the form $\widetilde{K}(k) \propto [(\omega\tau)^2+k^2]^{-1}$, i.e. a Yukawa-type kernel. This Fourier representation is not used in the subsequent calculations, but it is useful to highlight the effective range and structure of the induced interactions on the Ising side. This structure reveals how different dynamical regimes of the Rabi model appear on the Ising side. For weak coupling $g \ll \omega,\omega_0$, the long-range piece is negligible and the dynamics is dominated by the short-range stiffness. Conversely, in the ultra-strong and deep-strong coupling regimes $g\sim \omega$ or $g \gg \omega$, the ferromagnetic tail becomes long-ranged and sizable. This behavior encodes the crossover between perturbative dynamics and the strongly correlated regime, providing an explicit Ising-side diagnostic of the dynamical phases of the Rabi model.

---

## Round 1 · Referee Report · Anonymous (Referee 2) · 2025-9-3

Strengths
2) Full mathematical details which make following a complex set of arguments possible
Weaknesses
Report
Requested changes
1) The coherent states used in eqn 2 are not the usual ones, as is made clear in the appendix, they are a combination of the regular coherent states along with a particular state for the spin. This should be clarified in the main text.
2) It would be interesting to discuss in more detail exactly what form the coupling in the Ising model has. For example, how does K depend on distance for different coupling strengths in the Rabi model? Are there signatures there of the different dynamical regimes present at weak and strong coupling?
Recommendation
Publish (meets expectations and criteria for this Journal)

---

## Round 1 · Referee Report · Emanuele Dalla Torre (Referee 3) · 2025-10-1

Report
Before being able to make a recommendation, I would need the authors to address the following four major comments.
Major comments:
1. While intellectually stimulating, this work has limited applicability: one is generically interested in the real time dynamics, rather than in the imaginary one. Imaginary times is usually introduced to obtain finite temperature or ground state expectation values, but in this case one does not compute overlaps between coherent states. Can the author propose at least one scenario in which the quantity that they compute has a physical meaning? (The text below Eq. (28) suggests that in principle one can use this approach to compute the partition function of the Rabi model at finite temperatures, but this is calculation is not carried over in practice and translated into physical observables).
2. The (imaginary) time evolution of the Rabi model can be computed by truncating the Hilbert space of the bosonic mode. Can the authors demonstrate explicitly that the present mode is more efficient in terms of classical computational resources?
3. The method allows one to compute the overlap with respect to the eigenstates of the operators b, which are not the operators of the original model (a). The authors state (page 4) that one can go from one to other through “appropriate linear combinations”. Can the author explain this point in detail?
4. Section 5 seems to me pretentious: The Ising model under present consideration is a fine tuned one with a very specific type of interactions. Why would anyone encounter such a model in real life, except through the present derivation? Alternatively, can one learn something generic about Ising models through this mapping?
5. The Rabi model is known to undergo a quantum phase transition (in the appropriate limit). Is the present method applicable to this limit? How does the phase transition affect the corresponding Ising model? Also, does the classical model at hand have a phase transition? (It probably depends on the scaling of the long range correlations)
Minor comments:
6. Fig. 1 – add labels on the subplots to help the reader understand the difference between the different plots without reading the caption.
7. Style: the text is at some times sloppy and can be improved using common LLM tools.
8. Typos: Page 6: haven’t --> have not
9. The mapping reminds me of the Coulomb gas formalism used to study imaginary time field theories. See for example https://en.wikipedia.org/wiki/Coulomb_gas . Perhaps there is a relation between these approaches (I’m not sure)?
Recommendation
Ask for major revision

---

## Round 2 · Referee Report · Emanuele Dalla Torre (Referee 3) · 2025-11-5

Report

The authors addressed all my comments in a satisfactory way and I now support the publication of the manuscript in its present form.

Recommendation

Publish (meets expectations and criteria for this Journal)

---

## Round 2 · Author Response

Dear Editor,

We thank you and the referees for the careful reading of our manuscript and for the constructive comments. Below we reproduce the referee comments and our corresponding responses. All the requested corrections and clarifications have been implemented in the revised version of the manuscript.

Sincerely, Bruno Scheihing-Hitschfeld and Néstor Sepúlveda

Report 1

We appreciate the referee’s overall positive evaluation and the suggestion to emphasize more clearly that the Trotterization in our analysis is carried out in real time.

In the revised version, we have added sentences in the abstract and introduction highlighting this fact, as well as a further sentence in the introduction to contrast it with the Trotterization in imaginary time that we discuss later on in the manuscript. Furthermore, we have added a paragraph at the end of section 3 to make it clear that everything discussed thus far are real-time results. We have modified the section titles to clarify the scope of each section as well.

We also thank the referee for pointing out to us the limit $\omega/\omega_0 \to 0$ and the associated transition to a superradiant phase. While a full analysis of this regime is beyond the present scope of our work, we have included a paragraph in the conclusions mentioning it as an outlook for future work.

1) Referee's comment: When reading the article, I was confused by the definition of coherent states and amplitudes, in particular Eq. 2 and Eq. 3. For Eq. 2, it is clear that $|\alpha\rangle$ is the regular coherent state, satisfying the usual condition $a |\alpha\rangle = \alpha |\alpha\rangle$ (as stated before Eq. 2). However, in this case $A_{\beta\alpha}$ of Eq. 2 would be a spin operator. Instead, a few lines below $A_{\beta\alpha}$ is called "amplitude" and amplitudes are usually c-numbers. Even more confusing was for me the following section 2, as $|\alpha\rangle$ now indicates a different type of coherent state. The Authors actually note at the end of Section 1 that they use the coherent states of $b$ (and not of $a$) but I just assumed that they take them in the simple form $|\alpha\rangle|\pm\rangle$, where the spin states are eigenstates of $\sigma_x$. This choice, I would say, is the most intuitive one since $b = a\sigma_x$. However, also in Eq. 3 the spin degrees of freedom do not appear explicitly.

As it turns out, the form of the coherent states is different from my initial guess, and is only given in Appendices A and B. To avoid misunderstandings, I think it is better to define the coherent states of $b$ explicitly in the main text, before introducing the amplitudes. Then, one can appreciate immediately that they are entangled states between spin and cavity with an additional subscript $\pm$ (distinguishing $|\alpha\rangle_\pm$ from the usual $|\alpha\rangle$), which is later omitted.

Response: This is a point that all three referees raised in some form, which we clearly should have explained in more detail. To clarify this point, we made the following modifications: i) To clarify our notation, we added explicit spin labels in Eq. (2), and denoted the usual coherent states (eigenstates of $a$) with a subscript $c$'', i.e., $|\alpha\rangle_c$. Given their widespread use in our work, we reserve $|\alpha\rangle$ for the eigenstates of $b$. While we could move on directly to the latter (generalized) coherent states, we find it natural to first mention the traditional coherent states. ii) To clarify that our results will \textit{not} be given in terms of Eq. (2), we have added a paragraph after Eq. (2):In this work, we will derive a formula to calculate a family of directly related amplitudes -- which may in fact be used to reconstruct Eq. (2) -- without solving a differential equation. We will do so in terms of a set of generalized coherent states, defined as eigenstates of spin-dependent annihilation operators $b \equiv \sigma^x a$ and a parity operator $\Pi$, introduced later in the text. For brevity, we will denote these generalized states simply by $|\alpha \rangle$ by working in a fixed parity sector, with the understanding that they differ from the usual coherent states of $a$ (denoted, as above, by $|\alpha \rangle_c$). Explicit formulas and further details of these states are provided in Appendix A, where the parity operator $\Pi$ and the photon-atom entanglement structure of these states are discussed at length.'' The following paragraph now reads The formula we will derive is nothing else than the partition function of an Ising model''. iii) We have added additional text at the end of the last paragraph of the introduction. AfterThe eigenstates of $a$ and $b$ are nonetheless related to each other by taking appropriate linear combinations'', we now give further discussion showing how this takes place.

2) Referee comment: I was confused by the meaning of $\delta_{\ell,j+1}$ in Eq. 15 (perhaps because of the concomitant presence of $\delta_n$). Maybe it is not completely useless to remind the reader that $\delta_{\ell,j+1}$ is a Kronecker delta.

Response: We addressed this point by adding the following clarification in the text:
``Note that $\delta_{\ell,j+1}$ in Eq. (18) denotes a Kronecker delta, ensuring that the logarithmic term contributes only when $\ell=j+1$.''

3) Referee comment: Equation A.1 does not coincide with Eq. 1, since $\sigma^+\sigma^-$ is not equal to $\sigma_z$. The relation on line 284 is wrong and should be corrected. Perhaps it is better to directly use $\sigma_z$ in Eq. A.1, like in Eq. 1.

Response: We thank the referee for pointing out this mistake. Indeed, $\sigma^+\sigma^-$ is not equal to $\sigma_z$. We have corrected Eq. (A1) by explicitly writing it in terms of $\sigma_z$, in agreement with Eq. (1). In addition, the corresponding sentence on line 284 has been modified to be consistent with this correction.

4) Referee comment: I noticed these minor misprints:
- there is a space missing on line 283, before ``hbar=1''
- on line 325, evolution $\to$ evolve?
- $|0\rangle$ is missing on the rightmost side of Eq. B.4

Response: We thank the referee for carefully pointing out these misprints. We have corrected all three: (i) we added the missing space before $\hbar=1$'' on line 283, (ii) we changedevolution'' to ``evolve'' on line 325, and (iii) we included the missing $|0\rangle$ on the rightmost side of Eq. (B4) (which is now Eq. (B.6)).

Report 2

We thank the referee for the positive assessment under “Strengths” and for pointing out that the analysis of the resulting Ising model was somewhat superficial. We have addressed this by expanding the discussion of the Ising couplings and their relation to different dynamical regimes of the Rabi model.

1) Referee comment:
The coherent states used in Eq. 2 are not the usual ones, as is made clear in the appendix, they are a combination of the regular coherent states along with a particular state for the spin. This should be clarified in the main text.

Response: This is a point that all three referees raised in some form, which we clearly should have explained in more detail. To clarify this point, we made the following modifications: i) To clarify our notation, we added explicit spin labels in Eq. (2), and denoted the usual coherent states (eigenstates of $a$) with a subscript $c$'', i.e., $|\alpha\rangle_c$. Given their widespread use in our work, we reserve $|\alpha\rangle$ for the eigenstates of $b$. While we could move on directly to the latter (generalized) coherent states, we find it natural to first mention the traditional coherent states. ii) To clarify that our results will \textit{not} be given in terms of Eq. (2), we have added a paragraph after Eq. (2):In this work, we will derive a formula to calculate a family of directly related amplitudes -- which may in fact be used to reconstruct Eq. (2) -- without solving a differential equation. We will do so in terms of a set of generalized coherent states, defined as eigenstates of spin-dependent annihilation operators $b \equiv \sigma^x a$ and a parity operator $\Pi$, introduced later in the text. For brevity, we will denote these generalized states simply by $|\alpha \rangle$ by working in a fixed parity sector, with the understanding that they differ from the usual coherent states of $a$ (denoted, as above, by $|\alpha \rangle_c$). Explicit formulas and further details of these states are provided in Appendix A, where the parity operator $\Pi$ and the photon-atom entanglement structure of these states are discussed at length.'' The following paragraph is now a single sentence The formula we will derive is nothing else than the partition function of an Ising model''. iii) We have added additional text at the end of the last paragraph of the introduction. AfterThe eigenstates of $a$ and $b$ are nonetheless related to each other by taking appropriate linear combinations'', we now give further discussion showing how this takes place.

2) Referee comment: It would be interesting to discuss in more detail exactly what form the coupling in the Ising model has. For example, how does $K$ depend on distance for different coupling strengths in the Rabi model? Are there signatures there of the different dynamical regimes present at weak and strong coupling?

Response: We have expanded the discussion by adding two paragraphs in Section 4 [after Eq. (25)], which explain more explicitly the form of the Ising couplings and how they reflect the different dynamical regimes of the Rabi model.

Report 3

We thank the referee for the careful reading of our manuscript and for the constructive comments. Below we address each of the points in detail, indicating the corresponding changes introduced in the revised version.

1) Referee comment: While intellectually stimulating, this work has limited applicability: one is generically interested in the real time dynamics, rather than in the imaginary one. Imaginary times is usually introduced to obtain finite temperature or ground state expectation values, but in this case one does not compute overlaps between coherent states. Can the author propose at least one scenario in which the quantity that they compute has a physical meaning?

Response: We are very grateful that the referee raised this comment, as it made us realize that our manuscript was not sufficiently clear in separating the real-time discussion, derivation and results (Sections 2 and 3), from the imaginary-time applications we discuss later on (Section 4, former Sections 4 and 5). In fact, what we consider to be our main results are contained in Sections 2 and 3. We have now clearly emphasized that this is so by adding sentences "This is our first/second main result" following Eqs. (17) and (20). These are real-time results, which allow one to directly calculate the real-time amplitudes in the Rabi model. This is what is shown in Figure 1. In the revised version, we have added sentences in the abstract and introduction highlighting this fact, as well as a further sentence in the introduction to contrast it with the Trotterization in imaginary time that we discuss later on in the manuscript. Furthermore, we have added a paragraph at the end of section 3 to make it clear that everything discussed thus far are real-time results. We have modified the section titles to clarify the scope of each section as well.

2) Referee comment: The (imaginary) time evolution of the Rabi model can be computed by truncating the Hilbert space of the bosonic mode. Can the authors demonstrate explicitly that the present mode is more efficient in terms of classical computational resources?

Response: We addressed this by discussing a direct comparison with the cost of calculating the amplitudes using a Hilbert-space truncation. In Section 2, we added (next-to-last paragraph)
In practice, calculating these amplitudes directly in the Rabi model becomes increasingly expensive at large values of $\alpha$ and $\beta$, since one must numerically represent a Hilbert space large enough to accommodate the coherent states $\ket{\alpha}$ and $\ket{\beta}$. Exact diagonalization methods require a cutoff $N_{\text{max}}$ on the photon number basis, with $N_{\text{max}}$ typically scaling as $|\alpha|^2$ to achieve convergence, because the Poissonian weight $p_N = e^{-|\alpha|^2} |\alpha|^{2N}/N!$ extends up to $N \approx |\alpha|^2$. By contrast, our mapping avoids this: it relies on the partition function of an associated Ising model and thus does not require explicit overlaps with a truncated bosonic basis. In our approach, the cost is completely contained within the time evolution. In particular, the Trotter discretization converges with a number of time slices $n$ that scales linearly with $|\alpha|$ (see Appendix B.2. As we discuss in the next section in terms of an expansion in the number ofdomain walls'' of the partition function, and in more generally in Appendix B.2, this can lead to more efficient calculations than the Hilbert space truncation.''

In Section 3, we added `` It is worth emphasizing that in going from Eq. (17) to Eq. (20), $m$ is always smaller than $n$, meaning that the information encoded in the first $n$ terms of Eq. (17) is contained within the first $n$ terms of Eq. (20). Taken together with the estimate in Appendix B.2, this suggests that to stay below an error tolerance $\varepsilon$, one needs $m \gtrsim C_1 g \omega |\alpha| t^2/\varepsilon$, scaling linearly with $\alpha$. However, inspecting Eq. (20), we see that this might be an overly conservative estimate, as the result is organized as a power series in $\omega_0 t$, while the sensitivity to $\alpha,\beta$ is purely contained in the integral moments $F_m$. In particular, a scan over the values of $\omega_0$ could be done (up to a given tolerance fixed by the value of $\omega_0 t$) at significantly lower cost than with a direct diagonalization approach if one calculates the functions $F_m$ in advance.''

The derivation in Appendix B.2 shows how these cost estimates are obtained, and it discusses both the real- and the imaginary-time formulations.

3) Referee comment: The method allows one to compute the overlap with respect to the eigenstates of the operators $b$, which are not the operators of the original model ($a$). The authors state (page 4) that one can go from one to the other through “appropriate linear combinations”. Can the author explain this point in detail?

Response: This is a point that all three referees raised in some form, which we clearly should have explained in more detail. To address this comment, we have substantially expanded the discussion of the relation between $a$ and $b$: i) To clarify our notation, we added explicit spin labels in Eq. (2), and denoted the usual coherent states (eigenstates of $a$) with a subscript $c$'', i.e., $|\alpha\rangle_c$. Given their widespread use in our work, we reserve $|\alpha\rangle$ for the eigenstates of $b$. While we could move on directly to the latter (generalized) coherent states, we find it natural to first mention the traditional coherent states. ii) To clarify that our results will \textit{not} be given in terms of Eq. (2), we have added a paragraph after Eq. (2):In this work, we will derive a formula to calculate a family of directly related amplitudes -- which may in fact be used to reconstruct Eq. (2) -- without solving a differential equation. We will do so in terms of a set of generalized coherent states, defined as eigenstates of spin-dependent annihilation operators $b \equiv \sigma^x a$ and a parity operator $\Pi$, introduced later in the text. For brevity, we will denote these generalized states simply by $|\alpha \rangle$ by working in a fixed parity sector, with the understanding that they differ from the usual coherent states of $a$ (denoted, as above, by $|\alpha \rangle_c$). Explicit formulas and further details of these states are provided in Appendix A, where the parity operator $\Pi$ and the photon-atom entanglement structure of these states are discussed at length.'' The following paragraph now reads The formula we will derive is nothing else than the partition function of an Ising model''. iii) We have added additional text at the end of the last paragraph of the introduction. AfterThe eigenstates of $a$ and $b$ are nonetheless related to each other by taking appropriate linear combinations'', we now give further discussion showing how this takes place.

4) Referee comment: Section 5 seems to me pretentious: The Ising model under present consideration is a fine tuned one with a very specific type of interactions. Why would anyone encounter such a model in real life, except through the present derivation? Alternatively, can one learn something generic about Ising models through this mapping?

Response: To clarify the scope of this section, we have changed the title to "What the Rabi model can do for this Ising model" (and also for section 4). We have also grouped Sections 4 and 5 together into a new Section 4, to make it clear that this discussion applies to imaginary time, and is separate from the real-time discussion in Sections 2 and 3, which contain real-time results.

To illustrate a potential application of this section, we have added a paragraph at the beginning of this section: "Although the Ising model obtained here is a specific instance thereof, with couplings determined by those of the Rabi model, we emphasize that partition functions like the ones we just encountered do appear in other contexts. For example, they share structural features with long-range interacting spin systems that naturally arise in cavity and waveguide QED [Rev.Mod.Phys. 90 (2018) 3, 031002]. The mapping therefore has relevance beyond a formal exercise, as it highlights a class of interactions of direct physical interest."

We hope that these two changes make the scope and purpose of this section clear, and that it does not strike readers as pretentious.

5) Referee comment: The Rabi model is known to undergo a quantum phase transition (in the appropriate limit). Is the present method applicable to this limit? How does the phase transition affect the corresponding Ising model? Also, does the classical model at hand have a phase transition?

Response: This is a very relevant point, which was also raised in the first report we received. However, since the main point of our work is to establish a correspondence with the real-time amplitudes, we consider the exploration of these questions beyond the scope of the present manuscript. We hope that the modifications we have made will make the scope of our presentation clear.

We added a paragraph to the Outlook discussing our expectations:
``In principle, the present method could also be applied to the limit $\omega/\omega_0 \to 0$, where the Rabi model undergoes a superradiant phase transition characterized by the spontaneous buildup of a macroscopic photon field [PRL 115(18), 180404 (2015)] (see also [PRA 85(4), 043821 (2012)]). Within our Ising correspondence, this regime maps onto a situation where the effective long-range kernel $K^E_{j,\ell}$ strongly suppresses domain walls, favoring globally ordered spin configurations. We expect that in this spin representation, the onset of the superradiant phase would corresponds to the development of a nonzero magnetization --- analogous to having a finite coherent amplitude in the bosonic description. Speculatively, the exponential decay of the kernel would suppress domain walls sufficiently to stabilize ordered phases such that it would mirror the emergence of superradiance in the Rabi model as $\omega/\omega_0 \to 0$. Although this lies beyond our present scope, our Ising formulation offers a framework to investigate this superradiant limit from a complementary angle.''

Minor comments:

6) Referee comment: Fig. 1 – add labels on the subplots to help the reader understand the difference between the different plots without reading the caption.

Response:
We thank the referee for this useful suggestion. We have added labels to the subplots in Fig. 1 in the revised version.

7) Referee comment: Style: the text is at some times sloppy and can be improved using common LLM tools.

Response: We have reviewed and revised the text to improve clarity and conciseness throughout the manuscript.

8) Referee comment: Typos: Page 6: haven’t $\to$ have not.

Response: We have done so. We thank the referee for spotting this.

9) Referee comment: The mapping reminds me of the Coulomb gas formalism used to study imaginary time field theories. Perhaps there is a relation between these approaches?

Response: This is an interesting point. After being prompted by the referee, we considered this possibility, but we did not arrive at any sharp conclusion beyond the fact that the interactions of the spin chain under consideration are non-local.

---

## Round 2 · List of Changes

Changes to the Introduction and Abstract:

1) Emphasizing the use of Trotter’s formula in real time. The revised manuscript explicitly states that the correspondence is obtained through Trotter’s formula applied to the real‐time evolution of the Rabi model. It also adds that the same construction naturally extends to the imaginary‐time (Euclidean) formulation. The new text is at the beginning and end of the third paragraph of the introduction. We also added a remark at the end of the second sentence of the abstract to highlight the real-time aspect nature of the Trotterization.

2) Object of study and coherent‐state notation. In both the previous and the revised versions the calculation uses coherent states associated with the spin–dependent operator $b=\sigma^x a$, but in the original version this point was not clearly distinguished from the standard photon coherent states. The resubmission clarifies from the start that all calculations are carried out within a fixed parity sector of $\Pi$, using generalized coherent states $\ket{\alpha}$ defined by $b\ket{\alpha}=\alpha\ket{\alpha}$, and it makes explicit the linear relations between these $b$–coherent states and the photon–spin product states $\ket{\pm,\alpha}_c$. Thus, both versions employ the same basis, but the revised text resolves the ambiguity that existed previously. We now make it clear in the first sentence of the abstract that we study ``a class of generalized coherent states''. Furthermore, we have added a paragraph after Eq. (2) to further clarify this, and expanded the discussion after Eq. (5) to explain how the states in each basis can be written in terms of each other. Additional details on the relation between the eigenstates of $a$ and $b$ are now included in Appendix~A.

Changes to the section ``Trotter’s formula and the (real-time) correspondence’’:

1) Section title and emphasis. The main visible change is the addition of “real-time’’ to the title, making explicit that this section deals with real-time quantum dynamics rather than equilibrium or Euclidean formulations.

2) Link to the main amplitude. The revised text (above the current Eq.~(16)) explicitly connects this section to Eq.~(8), clearly identifying it as the central observable of the paper. In the earlier version, this link was only implicit.

3) Derivation of the Ising correspondence. The overall structure of the mapping remains unchanged, but the resubmission introduces two clarifications:
(i) it explicitly identifies $\delta_{\ell,j+1}$ in Eq.~(18) as a Kronecker delta, and
(ii) it designates Eq.~(17) as \emph{“our first main result,”} emphasizing its central role within the paper.

4) Computational-scaling analysis (new). The next-to-last paragraph of this section presents a new discussion and makes a first reference to the new Appendix~B.2, comparing the computational scaling of two approaches: direct diagonalization and the number of steps needed in the Trotter–Ising formulation. It shows that the former requires a photon-number cutoff that scales as $N_{\text{max}}!\sim|\alpha|^2$, whereas the latter scales only linearly with $|\alpha|$.

5) Improved continuity. The revised version ends by linking naturally to the imaginary-time formulation in Sec.~4, improving the narrative flow. The earlier version concluded with a brief, less structured forward reference.

Changes to the section ``The continuum limit of the Ising model as a perturbative expansion in $\omega_0$''

1) Identification of main result. The revised version explicitly labels Eq.~(20) as the \emph{second main result}, highlighting its importance. The original version introduced this equation as part of the derivation without drawing attention to its central role.

2) Figure clarification. In the new version of Figure 1, additional information has been incorporated to make the presentation self-contained. At the top of each column, the values of the generalized coherent states $\alpha$ and $\beta$ used to generate the three plots in that column are explicitly indicated. Likewise, at the beginning of each row, the coupling ratios $g/\omega$ corresponding to the plots in that row are shown. Furthermore, the caption now specifies that all panels were computed with $\omega_0/\omega = 0.3$. These additions make the figure clearer.

3) Scaling and computational efficiency. A new paragraph (next-to-last of this section) relates the number of terms in the expansion to computational cost, noting that $m<n$ and that $m$ scales linearly with $|\alpha|$. This quantitative connection to efficiency was absent in the earlier version.

4) Emphasis on real-time formulation. The revised version ends with a new paragraph explicitly stating that all results refer to real-time dynamics, distinguishing them from imaginary-time approaches. This clarification was not present before. This new text also connects the continuum expansion back to the real-time Ising correspondence and forward to the imaginary-time discussion, whereas the earlier version treated this section as a standalone derivation.

Changes to original sections 4 and 5, now unified into section 4 ``Applications in imaginary time''

1) Scope and organization. The original text had two distinct sections—“What an Ising model can do for the Rabi model” and “What the Rabi model can do for an Ising model”. In the new version, these are unified into a single section, “Applications in imaginary time,” with two subsections reflecting the same bidirectional logic. We also changed an Ising model'' tothis Ising model'' to be clear that we are referring to the specific form of spin-spin interactions that appear as a result of the mapping.

2) Imaginary-time focus. Although both versions use Euclidean expressions, the revised one explicitly frames the discussion as imaginary-time applications of the correspondence, connecting it to finite-temperature and statistical-physics contexts. The earlier version implied this but did not present it as the main organizing idea.

3) Analysis of the Ising couplings. After Eq. (25), the new text includes a detailed breakdown of the kernel $K^E_{j,\ell}$ into a nearest-neighbor term and an exponentially decaying long-range tail, noting that the latter becomes a Yukawa-type kernel in the continuum limit.

4) Regime correspondence. In the paragraph that precedes the one containing Eq. (26), the revised version introduces an explicit mapping between dynamical regimes: weak coupling corresponds to short-range Ising interactions, while ultra- and deep-strong coupling lead to long-range ferromagnetic behavior that suppresses domain walls. This diagnostic link between Rabi and Ising dynamics is newly articulated.

5) Physical interpretation and broader context. The new version adds discussion between Eqs. (28) and (29) emphasizing that the Euclidean amplitudes yield the finite-temperature Rabi partition function and can in principle be used to compute physical observables such as linear-response functions. In the starting paragraph of section 4.2, it also notes that similar long-range Ising-like models appear in other quantum-optical settings (e.g., cavity or waveguide QED), extending the relevance of the mapping. These physical connections were not present in the earlier version of this section.

Changes to the conclusions & outlook

1) Scope and framing. The original conclusion connects the Ising–Rabi correspondence to dualities and quantum-optical analogies. The revised version also highlights how the physical regimes of the Rabi model are captured by the mapping, emphasizing how the Ising couplings encode different dynamical behaviors of the Rabi model. In particular, we have added a new explanatory sentence at the end of the first paragraph linking the mapping to the Rabi dynamical regimes: “This perspective highlights the broader relevance of our approach and clarifies how the effective Ising couplings reflect the different dynamical regimes of the Rabi model.” This conceptual connection was absent previously.

2) Clarification of the coherent states. The first sentence now explicitly refers to generalized coherent states, clarifying that these are the same $b=\sigma^x a$ eigenstates used before but now defined precisely and distinguished from standard photon coherent states.

3) Superradiant phase transition. The new version adds a discussion of the superradiant limit $\omega/\omega_0 \to 0$ in the next-to-last paragraph, where we discuss our expectation that in this regime the emergence of a macroscopic photon field is accounted for by the suppression of domain walls in the long-range Ising kernel.

---

## Editorial Decision

published